# NPC1 controls TGFBR1 stability in a cholesterol transport-independent manner and promotes hepatocellular carcinoma progression

Shuangyan Li [1,2,6], Lishan Yan[2,6], Chaoying Li [2], Lijuan Lou[2], Fengjiao Cui[2,3], Xiao Yang[2], Fuchu He [2,4,5] ✉ & Ying Jiang [2,5] ✉

Niemann-Pick disease type C protein 1 (NPC1), classically associated with cholesterol transport and viral entry, has an emerging role in cancer biology. Here, we demonstrate that knockout of *Npc1* in hepatocytes attenuates hepatocellular carcinoma (HCC) progression in both DEN (diethylnitrosamine)-CCl$_4$ induced and MYC-driven HCC mouse models. Mechanistically, NPC1 significantly promotes HCC progression by modulating the TGF-β pathway, independent of its traditional role in cholesterol transport. We identify that the 692-854 amino acid region of NPC1's transmembrane domain is critical for its interaction with TGF-β receptor type-1 (TGFBR1). This interaction prevents the binding of SMAD7 and SMAD ubiquitylation regulatory factors (SMURFs) to TGFBR1, reducing TGFBR1 ubiquitylation and degradation, thus enhancing its stability. Notably, the NPC1 (P691S) mutant, which is defective in cholesterol transport, still binds TGFBR1, underscoring a cholesterol-independent mechanism. These findings highlight a cholesterol transport-independent mechanism by which NPC1 contributes to the stability of TGFBR1 in HCC and suggest potential therapeutic strategies targeting NPC1 for HCC treatment.

Liver cancer presents a significant global health challenge, with an increasing incidence worldwide[1–3]. It ranks as the sixth most common malignancy globally, with its mortality rate being the third highest among cancers[4,5]. Hepatocellular carcinoma (HCC) is the most prevalent form of primary liver cancer, accounting for approximately 90% of all liver cancer cases[1]. The treatment of HCC faces hurdles such as a high recurrence rate and a limited survival period, with existing clinical therapies yielding suboptimal results. Therefore, there is a pressing need for in-depth research into the molecular mechanisms driving HCC progression, which could lead to the discovery of innovative and

effective diagnostic biomarkers and drug targets, offering avenues for treatment.

Previous research has indicated that cholesterol homeostasis is significantly disrupted in HCC[6]. NPC1, a protein commonly associated with cholesterol transport, has been identified as having a high prognostic risk score in HCC[6,7]. NPC1 is a large protein, that spans the membrane 13 times and is predominantly localized in the late endosome/lysosome (LE/Ly) membrane. Diseases related to NPC1 include Niemann-Pick disease type C1, a rare autosomal recessive genetic disorder. Dysfunctional NPC1, resulting from mutations in the *NPC1*

[1]School of Life Sciences, Tsinghua University, Beijing, China. [2]State Key Laboratory of Medical Proteomics, National Center for Protein Sciences (Beijing), Beijing Proteome Research Center, Beijing Institute of Lifeomics, Beijing, China. [3]School of Basic Medicine, Qingdao University, Qingdao, China. [4]Research Unit of Proteomics Dirven Cancer Precision Medicine, Chinese Academy of Medical Sciences, Beijing, China. [5]Anhui Medical University, Hefei, China. [6]These authors contributed equally: Shuangyan Li, Lishan Yan. ✉e-mail: hefc@bmi.ac.cn; jiangying@ncpsb.org.cn

gene, leads to excessive accumulation of intracellular lysosomal cholesterol. This pathogenesis is characterized by features such as hepatosplenomegaly, cognitive impairment, and progressive and disabling neurological symptoms[8–14]. Furthermore, mutations in *NPC1* have been identified as a risk factor for childhood and adult morbid obesity[15–17]. NPC1's role extends beyond genetic disorders, as it has been implicated in various viral infections. For instance, the Ebola virus spike glycoprotein (GP) binds specifically to NPC1 in the LE/Ly, triggering cellular infection[18–21]. Additionally, NPC1 has been associated with other viral infections, including HIV-1[22], Chikungunya virus[23] and certain hepatoviruses[24]. Despite these associations, the exact role of NPC1 in the context of HCC remains unclear, warranting further investigation.

In this work, we elucidate the critical roles of NPC1 in the development and progression of HCC using knockout (KO) mouse models. Interestingly, we find that NPC1 promotes the TGF-β pathway by stabilizing the protein TGFBR1, independent of its role in cholesterol transport. This function positions NPC1 as a facilitator of tumor progression and metastasis in HCC. Furthermore, we reveal that NPC1 interacts with TGFBR1 and impedes the binding between TGFBR1 and the SMAD7/SMURFs complex, thereby reducing the ubiquitylation of TGFBR1. These insights not only shed light on the molecular mechanisms underpinning HCC progression but also establish NPC1 as both a potential prognostic marker and therapeutic target for combating HCC.

## Results

### Up-regulation of NPC1 correlates with poor prognosis of HCC

We analyzed The Cancer Genome Atlas (TCGA) datasets and found that NPC1 was significantly upregulated in tumors compared with adjacent tissues in 52% (12 out of 23) of TCGA cancer types, including HCC (Fig. 1a and Supplementary Fig. 1a). We also found that both protein and mRNA levels of NPC1 were significantly elevated in HCC tissues compared with paired non-tumor tissues. Furthermore, NPC1 expression showed a significant increasing trend in correlation with the prognostically associated proteomic subtypes in HCC patients from the cohorts of Jiang et al.'s[6] and Gao et al.'s[25] (Fig. 1b, c and Supplementary Fig. 1b, c). Patients with high NPC1 expression had significantly worse overall survival (OS) and disease-free survival (DFS) than those with low NPC1 expression (Fig. 1d–i).

To further investigate the prognostic value of NPC1 in HCCs, we performed a tissue microarray (TMA)-based immunohistochemistry (IHC) study of NPC1 in HCC tumor and paired non-tumor liver tissues. High NPC1 expression was found to be significantly associated with poor prognosis in HCC patients (both OS and DFS), further suggesting that NPC1 plays a critical role in HCC (Fig. 1j-m and Supplementary Data 1).

### NPC1 promotes HCC progression

Evaluation of NPC1 function in HCC was performed both in vivo and in vitro. To investigate NPC1's role in HCC progression, we engineered PLC/PRF/5 cells with stable overexpression of NPC1, and HepG2 and MHCC-97H cells with stable knockdown of NPC1 (Fig. 2a and Supplementary Fig. 2a). Overexpression of NPC1 significantly increased the proliferation of PLC/PRF/5 cells (Supplementary Fig. 2b), while knockdown of NPC1 notably suppressed the proliferation of HepG2 and MHCC-97H cells (Supplementary Fig. 2c, d). Additionally, enhanced NPC1 expression substantially increased the migration and invasion abilities of PLC/PRF/5 cells (Fig. 2b, d), while silencing NPC1 significantly reduced these abilities (Fig. 2c, e and Supplementary Fig. 2e, f). To exclude potential off-target effects, we reintroduced NPC1 into NPC1-knockdown HepG2 and MHCC-97H cells (Fig. 2a and Supplementary Fig. 2a). Reintroduction of NPC1 almost completely restored the proliferation, migration, and invasion capacities in both cell lines (Fig. 2c, e and Supplementary Fig. 2c–f). Notably, the slight

reduction in proliferation rate (10%–15%, as shown in Supplementary Fig. 2g–i) could not account for the nearly 70% decrease in migration ability observed in serum-free medium.

To further explore NPC1's role in both tumor growth and metastasis, we utilized a subcutaneous tumor inoculation model alongside a mouse tail vein metastasis model. In the subcutaneous model, NPC1 knockdown led to a significant reduction in tumor size and weight, while reintroducing NPC1 effectively restored tumor growth (Supplementary Fig. 2j-m). In the tail vein metastasis model, NPC1 depletion resulted in a significant reduction in lung tumor metastases, further supporting NPC1's critical role in tumor metastasis (Fig. 2f–i). Collectively, these findings highlight NPC1 as a key promoter of HCC progression.

### NPC1 regulates the TGF-β pathway in a cholesterol transport-independent manner

To investigate the mechanism through which NPC1 promotes HCC progression, we performed proteomic analysis on PLC/PRF/5 and HepG2 cells following NPC1 knockdown. Differentially expressed proteins were identified, and pathway enrichment analysis revealed that NPC1 knockdown significantly inhibited the TGF-β pathway in HCC cells (Supplementary Fig. 3a, b). Notably, activation of the TGF-β pathway is a hallmark of S-III HCC, which is typically associated with a poor prognosis after first-line surgery[6]. Additionally, the TGF-β pathway is closely linked to epithelial-mesenchymal transition and cancer cell invasion and metastasis[26,27].

We further validated the influence of NPC1 on TGF-β signaling in HCC cells. In cells overexpressing NPC1, there was a significant increase in the protein levels of TGFBR1, p-SMAD2 and p-SMAD3 (Fig. 3a). Conversely, NPC1 knockdown led to a notable decrease in these protein levels (Fig. 3b and Supplementary Fig. 3c). Accordingly, NPC1 overexpression increased the mRNA levels of *MMP2*, *MMP9* and *COL5A3* in PLC/PRF/5 cells (Fig. 3c). In contrast, knockdown of NPC1 in HepG2 cells resulted in a downregulation of these genes (Fig. 3d). Furthermore, high expression of these target genes was significantly associated with reduced OS in HCC patients, as evidenced by data from Jiang et al.'s cohort (Supplementary Fig. 3d–f). Functionally, NPC1 knockdown reduced cell migration in HepG2 and MHCC-97H cells, independent of TGF-β1 treatment (Fig. 3e and Supplementary Fig. 3h). In cells overexpressing NPC1, migration was significantly increased regardless of TGF-β1 stimulation, although TGF-β1 treatment in the control group induced higher migration compared to NPC1-overexpressing cells without TGF-β1 (Supplementary Fig. 3g). These findings suggest that SMAD2/3 activation is a critical pathway regulated by NPC1 in promoting HCC metastasis.

Next, we explored whether NPC1's regulation of the TGF-β pathway depends on its cholesterol transport function. To assess the cholesterol transport activity, we employed Filipin III staining, a fluorescent probe that binds specifically to unesterified cholesterol in fixed cells[28]. As shown in Fig. 3g, NPC1 knockdown resulted in significant intracellular cholesterol accumulation. Re-expression of wild-type NPC1, which possesses intact cholesterol transport function, rescued this phenotype, reducing cholesterol accumulation to normal levels. However, the P691S mutant NPC1, which is known to lack cholesterol transport activity[29,30], did not reverse the cholesterol accumulation. Following reintroduction of NPC1 (wild-type or P691S mutant), we observed restored levels of TGFBR1 and p-SMAD2, as well as rescued cell migration capacity (Fig. 3f, h). We also explored the impact of cholesterol modulation on the TGF-β pathway using MβCD, a cyclic oligosaccharide commonly employed to deplete membrane cholesterol[31,32]. Our results showed that altering cholesterol levels with MβCD did not affect TGFBR1 expression or TGF-β pathway activity (Supplementary Fig. 4a, b). Additionally, treatment with U18666A, a specific NPC1 cholesterol transport inhibitor[33] did not affect TGFBR1 or p-SMAD2 levels at various concentrations (Supplementary Fig. 4c, d).

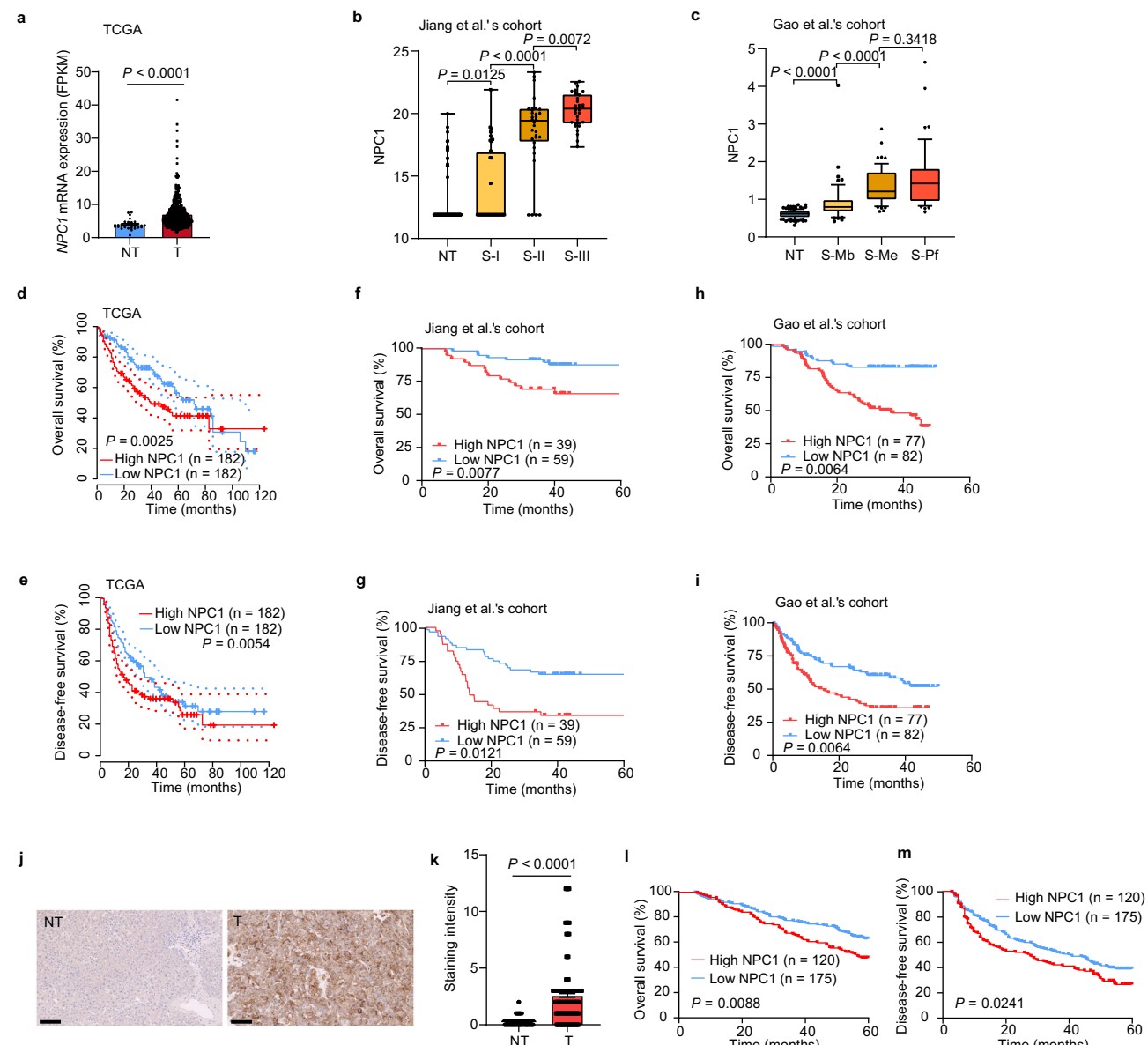

**Fig. 1 | Up-regulation of NPC1 correlates with poor prognosis of HCC.**
**a**–**c** Upregulation of NPC1 mRNA (**a**) or protein (**b**, **c**) in paired non-tumor tissues (NT) and tumor tissues (T) in TCGA datasets (**a**) (NT, $n = 32$; T, $n = 375$), Jiang et al.'s cohort (**b**) (NT, $n = 98$; S-I, $n = 36$; S-II, $n = 32$; S-III $n = 33$) and Gao et al.'s cohort (**c**) (NT, $n = 159$; S-Mb, $n = 55$; S-Me, $n = 57$; S-Pf, $n = 47$). **d**–**i** Kaplan–Meier overall survival (**d**, **f**, **h**) and disease-free survival (**e**, **g**, **i**) curves of individuals with high or low NPC1 expression in TCGA datasets (**d**, **e**), Jiang et al.'s cohort (**f**, **g**) and Gao et al.'s cohort (**h**, **i**). **j** Representative IHC staining of TMA with NPC1 antibodies in an independent cohort of HCC ($n = 295$ biologically independent samples); scale bars, 100 μm. **k** Staining intensity of NPC1 between NT and T samples from TMA ($n = 295$ biologically independent samples). **l**, **m** Kaplan–Meier overall survival (**l**) and disease-free survival (**m**) curves of individuals with high or low NPC1 expression. In the box plots, the middle bar represents the median, and the box represents the interquartile range; bars extend to 1.5× the interquartile range. Data are presented as the mean ± s.e.m. (**a**, **k**). Statistical significance was determined by Mann–Whitney U test (**a**–**c**, **k**) or log-rank test (**d**–**i**, **l**, **m**). Source data are provided as a Source Data file.

These results indicate that NPC1 regulates the TGF-β pathway in a manner independent of its cholesterol transport function.

## NPC1 increases protein stability of TGFBR1 and inhibits its ubiquitination

Subsequently, we observed that in PLC/PRF/5 cells with NPC1 over-expression, the protein levels of TGFBR1 were significantly elevated, while the corresponding mRNA levels remained unchanged (Figs. 3a, 4a). Similarly, in NPC1-knockdown HepG2 and PLC/PRF/5 cells, TGFBR1 protein levels were notably reduced, with no significant changes in mRNA expression (Figs. 3b, 4b and Supplementary Figs. 3c, 5a). Given that TGF-β receptors undergo constant internalization and recycling,

independent of ligand presence[34–36], our experiments indicated that TGF-β stimulation did not affect the half-life of the TGFBR1 protein in PLC/PRF/5 and HepG2 cells (Supplementary Fig. 5b, c). Cyclo-heximide (CHX) chase experiments demonstrated that NPC1 overexpression extended the half-life of TGFBR1 in PLC/PRF/5 cells (Fig. 4c), whereas NPC1 knockdown accelerated its degradation in HepG2 cells (Fig. 4d), underscoring NPC1's crucial role in stabilizing TGFBR1 protein levels. To elucidate the sustained dynamics of TGF-β signaling under NPC1 overexpression, we conducted time course experiments. In NPC1-overexpressing PLC/PRF/5 cells, SMAD2 phosphorylation was prolonged over a 24-hour period, whereas attenuation of the signal was observed in control cells (Supplementary Fig. 5e). This suggests that

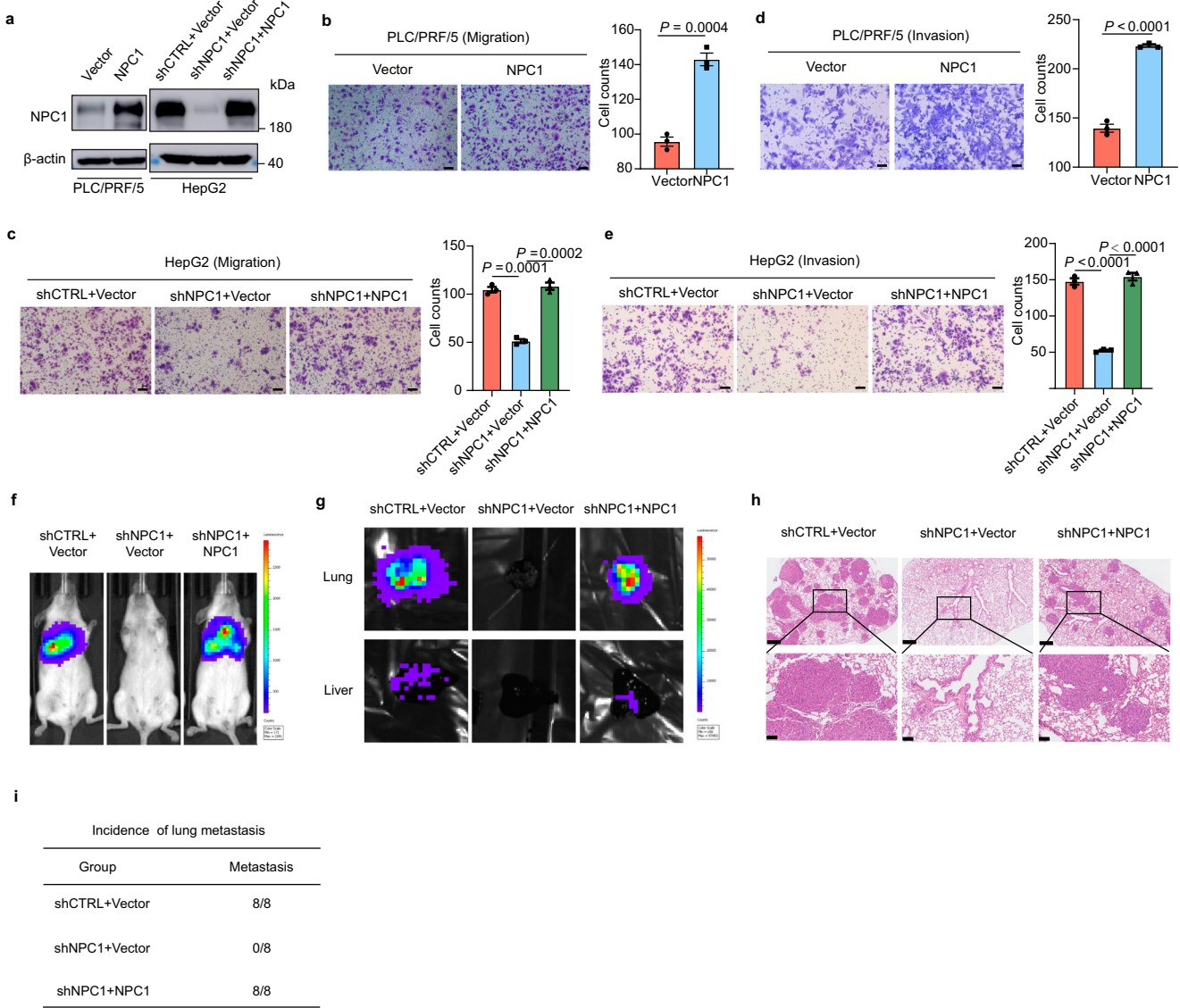

**Fig. 2 | NPC1 promotes HCC progression. a** Confirmation of NPC1 overexpression, NPC1 knockdown and re-expression in HCC cells. **b–e** Transwell assay to examine the effect of NPC1 on HCC cell migration (**b**, **c**) or invasion (**d**, **e**); scale bars, 100 μm. **f–i** 1×10⁶ Luciferase-expressing HCC cells (MHCC-97H) were injected into NOD SCID mice by tail vein. The mice were euthanized 8 weeks later by a cervical dislocation. Representative images of whole body luminescence monitoring of NOD SCID mice injected via tail vein with HCC cells 8 weeks after injection (**f**). Lung and liver tissues were isolated for analysis of IVIS imaging (**g**). Representative H&E staining images of lung tissues are shown; scale bars, 500 μm; insets: fivefold magnification; scale bars, 100 μm (**h**). The incidence of lung metastasis in mice (**i**). (*n* = 8 mice per group). Data are presented as the mean ± s.e.m. *n* = 3 (**b–e**) biologically independent samples (**b–e**). Statistical significance was determined by two-tailed unpaired Student's t-test (**b–e**). **a–e** Data were verified in three independent experiments. Source data are provided as a Source Data file.

NPC1 overexpression prolongs TGF-β signaling, likely by impairing receptor downregulation.

To further investigate the mechanism by which NPC1 regulates TGFBR1 degradation, we examined whether it was mediated by the proteasome or lysosome pathways. Treatment with the proteasome inhibitor MG132 led to a significant increase in TGFBR1 protein levels, while no such effect was observed with the lysosomal inhibitor NH₄Cl in HepG2 and PLC/PRF/5 cells with stable NPC1 knockdown (Fig. 4e and Supplementary Fig. 5d). Quantitative analyses of these immunoblots confirmed that proteasomal degradation is the predominant pathway for TGFBR1 degradation in the context of NPC1 knockdown (Fig. 4e and Supplementary Fig. 5d). These results suggest that NPC1 promotes TGF-β signaling by preventing proteasome-mediated degradation of TGFBR1.

Furthermore, we examined the ubiquitination of TGFBR1. In PLC/PRF/5 cells, NPC1 overexpression significantly reduced overall TGFBR1 polyubiquitination, whereas NPC1 knockdown in HepG2 cells had the opposite effect (Fig. 4f, g). Specifically, NPC1 decreased Lys 48-linked polyubiquitination, which is associated with proteasomal degradation, while Lys 63-linked polyubiquitination, typically involved in non-proteolytic functions, was unaffected (Fig. 4f, g). These data indicate that NPC1 stabilizes TGFBR1 by reducing its proteasomal degradation through inhibition of Lys 48-linked ubiquitination.

## NPC1 interacts with TGFBR1 and inhibits the binding of TGFBR1 with SMAD7/SMURFs

We explored the potential interaction between NPC1 and TGFBR1 through co-immunoprecipitation (Co-IP) assays, which confirmed that ectopic NPC1 binds to TGFBR1 in PLC/PRF/5 and HEK-293T cells (Fig. 5a and Supplementary Fig. 6a). This interaction was further validated endogenously in PLC/PRF/5 and HepG2 cells (Fig. 5b). To

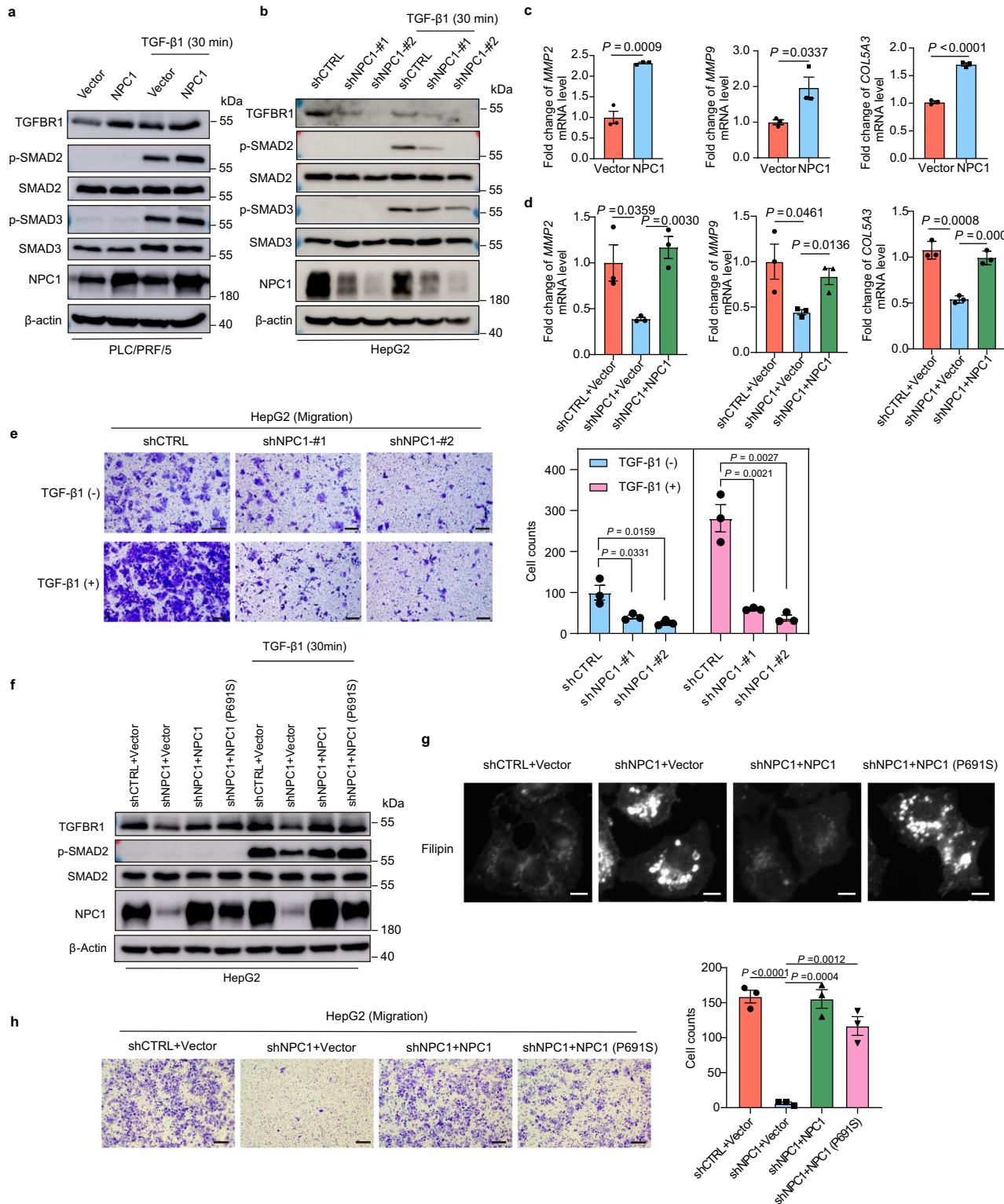

**Fig. 3 | NPC1 regulates the TGF-β pathway in a cholesterol transport-independent manner. a**, **b** Immunoblot analysis of TGFBR1, p-SMAD2, p-SMAD3, SMAD2, SMAD3 and NPC1 expression in PLC/PRF/5 cells with NPC1 stable over-expression (**a**) or HepG2 cells with NPC1 stable knockdown (**b**). **c**, **d** qPCR (n = 3 biological replicates) was used to examine the mRNA level of TGF-β target genes in NPC1-overexpression PLC/PRF/5 cells (**c**) or in NPC1-knockdown HepG2 cells with/without further overexpression of NPC1 (**d**). **e** Transwell assay was performed in NPC1-knockdown HepG2 cells with or without TGF-β1 (10 ng/mL) treatment; scale bars, 100 μm. **f** Immunoblot analysis of TGFBR1, p-SMAD2, SMAD2, and NPC1

expression in NPC1-knockdown HepG2 cells with further overexpression of NPC1 or NPC1 (P691S). **g** Cells related to (**f**) were fixed and stained with filipin to label free cholesterol accumulated in LE/Ly; scale bars, 10 μm. **h** Transwell assay was performed in cells related to (**f**); scale bars, 100 μm. Data are presented as the mean ± s.e.m. n = 3 (**c**, **d**, **e**, **h**) biologically independent samples. Statistical significance was determined by two-tailed unpaired Student's t-test (**c**, **d**, **e**, **h**). All experimental data were verified in three independent experiments. Source data are provided as a Source Data file.

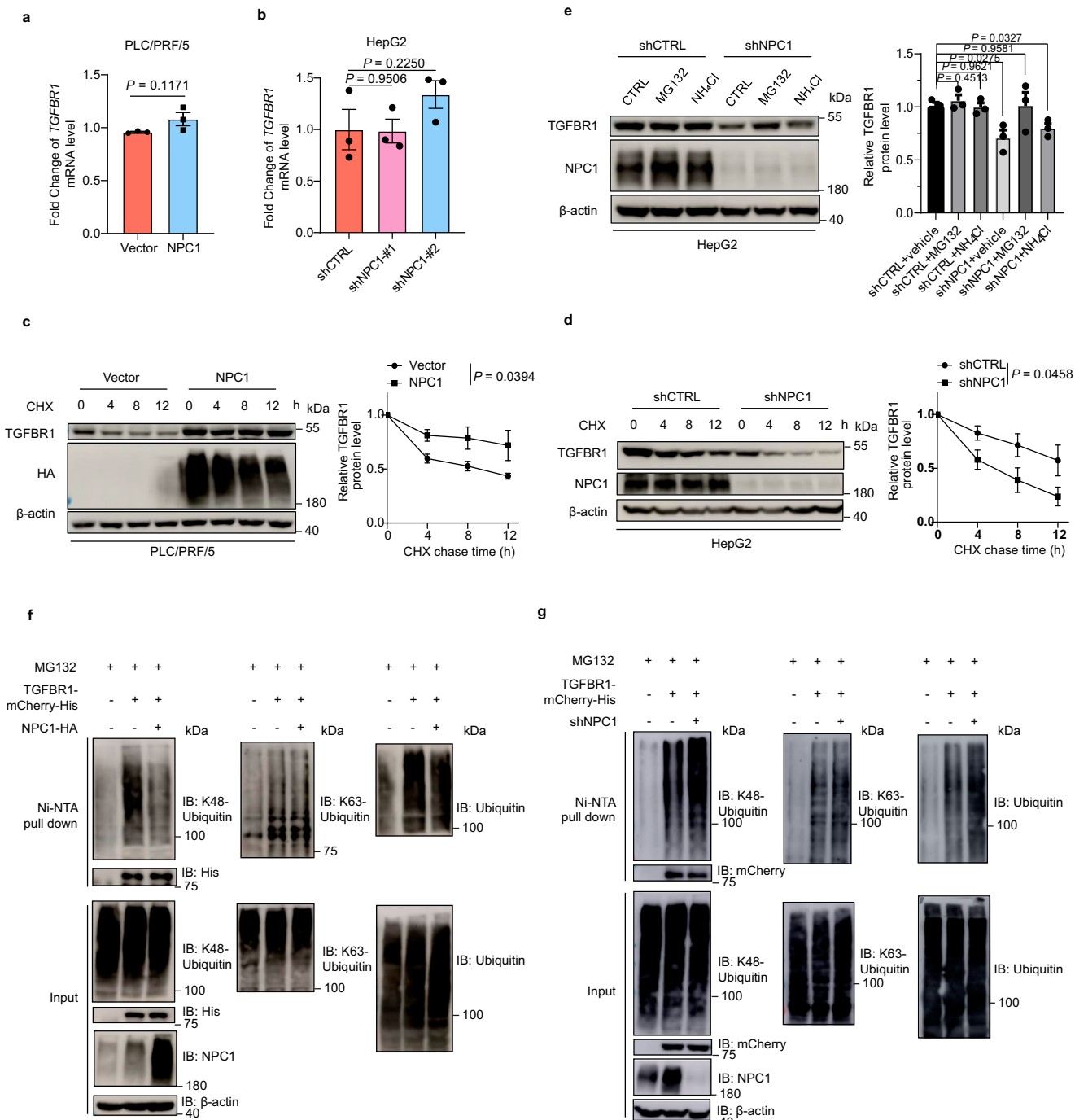

**Fig. 4 | NPC1 increases protein stability of TGFBR1 and inhibits its ubiquitination. a, b** qPCR analysis of NPC1 and TGFBR1 mRNA levels in NPC1-overexpression PLC/PRF/5 (**a**) or NPC1-knockdown HepG2 (**b**) cells. **c, d** PLC/PRF/5 cells with/without stable overexpression of NPC1 (**c**) or HepG2 cells with/without stable knockdown of NPC1 (**d**) were treated with CHX for indicated times and then analyzed by western blot. **e** HepG2 cells with or without stable knockdown of NPC1 were treated with vehicle, MG132 (10 μM), or NH₄Cl (10 mM) for 12 hours. Cell lysates were subjected to immunoblot with TGFBR1 or NPC1 antibody. **f, g** TGFBR1-mCherry-His stable overexpression PLC/PRF/5 (**f**) and HepG2 (**g**) cells with or without NPC1 overexpression (**f**) or knockdown (**g**) were pretreated with MG132 (10 μM) for 8 hours before collection. Then TGFBR1-mCherry-His was pulled down by Ni-NTA and immunoblotted with anti-K48-Ubiquitin, anti-K63-Ubiquitin and anti-Ubiquitin antibody. Data are presented as the mean ± s.e.m. *n* = 3 (**a**–**e**) biologically independent samples. Statistical significance was determined by two-tailed unpaired Student's t-test (**a**, **b**, **e**) or two-way analysis of variance (ANOVA) (**c**, **d**). All experimental data were verified in three independent experiments. Source data are provided as a Source Data file.

gain insights into the subcellular localization of the TGFBR1-NPC1 complex, we examined their co-localization patterns. Although prior studies reported that TGFBR1 undergoes sustained internalization and localizes to various cytoplasmic vesicles, including compartments marked by LAMP1 or caveolin-1[37–40], our results revealed that the TGFBR1-NPC1 complex predominantly colocalizes with the lysosomal marker LAMP1 rather than with caveolin-1 (Fig. 5c, Supplementary Fig. 6b-f). This suggests that, within HCC cells, the TGFBR1-NPC1 complex predominantly resides in lysosomes. Notably, neither overexpression nor knockdown of NPC1 altered the lysosomal localization of TGFBR1 (Fig. 5c, Supplementary Fig. 6h, j, b, g, i).

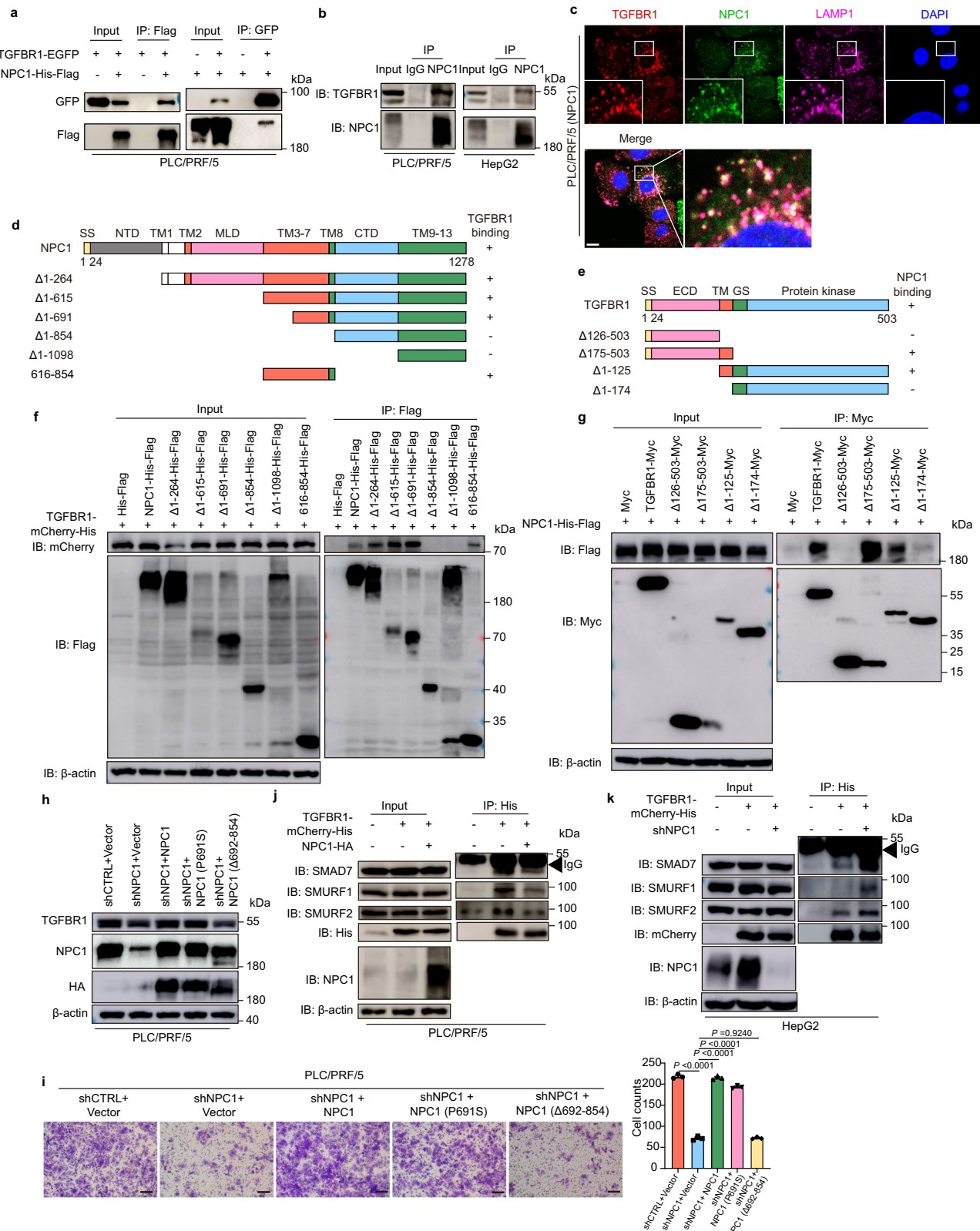

Further mapping of the NPC1-TGFBR1 interaction domain revealed that the amino acid (aa) 692-854 region of NPC1 is responsible for binding to TGFBR1 (Fig. 5d, f). The interaction surface of TGFBR1 with NPC1 was mapped to its transmembrane domain (Fig. 5e, g). Additionally, the NPC1 (P691S) mutant retained its ability to bind TGFBR1 (Supplementary Fig. 6m). Transwell assays were employed to examine the functional role of NPC1 truncations and mutants. In PLC/

PRF/5 cells with stable NPC1 knockdown, reintroduction of either wild-type NPC1 or NPC1 (P691S) restored TGFBR1 expression and cell migration, whereas reintroduction of NPC1 (Δ692-854) failed to rescue these phenotypes (Fig. 5h, i). Moreover, ectopic expression of wild-type NPC1 and NPC1 (P691S) significantly promoted cell migration, while NPC1 (Δ616-854) had no effect (Supplementary Fig. 6l), aligning with the observed effects of these mutants on TGFBR1 stability

**Fig. 5 | NPC1 interacts with TGFBR1 and inhibits the binding of TGFBR1 with SMAD7/SMURFs. a** The lysates of PLC/PRF/5 transfected with indicated constructs were subjected to immunoprecipitation with anti-Flag (or GFP) antibody. The immunoprecipitates were then immunoblotted with anti-GFP (or Flag) antibody. **b** PLC/PRF/5 and HepG2 cell lysates were subjected to immunoprecipitation with control IgG or anti-NPC1 antibodies. **c** PLC/PRF/5 cells stably overexpressing TGFBR1-mCherry-His and NPC1-HA were immunostained with antibodies against HA and LAMP1 to determine the colocalization among TGFBR1, NPC1 and LAMP1 in PLC/PRF/5 cells. Representative images from three independent experiments are shown; scale bars, 10 μm. **d, e** A schematic representation of NPC1 (**d**) or TGFBR1 (**e**) WT and deletion mutants. **f** PLC/PRF/5 cells stably overexpressing TGFBR1-mCherry-His were transfected with various plasmids encoding NPC1-His-Flag or NPC1 deletion mutants as indicated. Cell lysates were subjected to immunoprecipitation with anti-Flag magnetic beads and immunoblot with mCherry or Flag antibody. **g** PLC/PRF/5 cells were transfected with various combinations of plasmids encoding NPC1-His-Flag and TGFBR1-Myc or TGFBR1 deletion mutants as indicated. Cell lysates were subjected to immunoprecipitation with anti-Myc magnetic beads and immunoblot with Flag or Myc antibody. **h** Immunoblot analysis of TGFBR1 and NPC1 expression in NPC1-knockdown PLC/PRF/5 cells with further overexpression of NPC1, NPC1 (P691S), or NPC1 (Δ692-854). **i** Transwell assay was performed in cells related to (**h**); scale bars, 100 μm. **j, k** TGFBR1-mCherry-His stable overexpression PLC/PRF/5 (**j**) and HepG2 (**k**) cells with or without NPC1 overexpression (**j**) or knockdown (**k**) were subjected to immunoprecipitation with anti-His antibody. The lysates and immunoprecipitates were then blotted. Data are presented as the mean ± s.e.m. *n* = 3 (**i**) biologically independent samples. Statistical significance was determined by two-tailed unpaired Student's t-test (**i**). All experimental data were verified in three independent experiments. Source data are provided as a Source Data file.

(Supplementary Fig. 6k). These results suggest that the aa 692-854 region of NPC1 is essential for its role in promoting TGFBR1 stability and cell migration.

Given that SMAD7 recruits the E3 ubiquitin ligases SMURF1 and SMURF2 to TGFBR1, facilitating its ubiquitination and degradation[41–43], we hypothesized that NPC1 might suppress TGFBR1 ubiquitination by modulating its interaction with SMAD7 or SMURFs. Therefore, we speculated that NPC1 might suppress the ubiquitylation of TGFBR1 via modulating its interaction with SMAD7 or E3 enzymes. Co-IP assays revealed that NPC1 overexpression reduced the interaction between TGFBR1 and SMAD7, as well as with the E3 ligases SMURF1 and SMURF2, in PLC/PRF/5 cells (Fig. 5j). Conversely, NPC1 knockdown in HepG2 cells enhanced these interactions (Fig. 5k). Supporting these findings, colocalization studies of SMAD7-EGFP and TGFBR1-mCherry (Supplementary Fig. 7a, b) showed that in NPC1 knockdown cells, cytoplasmic localization of SMAD7-EGFP increased significantly and colocalized with TGFBR1-mCherry. Further analysis was conducted to determine whether NPC1 stabilizes TGFBR1 in a SMAD7-dependent manner. Knockdown of SMAD7 or inhibition of SMURF1/2 using siRNA rescued TGFBR1 protein levels in HCC cells with stable NPC1 knockdown (Supplementary Fig. 7c, d). These results imply that TGFBR1 degradation in NPC1 knockdown cells is driven, at least in part, by the increased recruitment of SMAD7 and SMURFs to TGFBR1. Collectively, these findings suggest that NPC1 protects TGFBR1 from proteasomal degradation by interacting with it and inhibiting its binding with SMAD7 and SMURFs.

## TGFBR1 is Crucial for NPC1-Mediated Promotion of HCC Progression

We examined the correlation between NPC1 and TGFBR1 protein levels in 286 pairs of human HCC tissue samples using immunohistochemistry (IHC). Our analysis revealed a significant positive correlation between NPC1 and TGFBR1 expression ($P < 0.0001$, $R^2 = 0.5028$) (Fig. 6a–c). To further explore this relationship, we performed immunofluorescence staining on HCC tissues with antibodies specific to NPC1 and TGFBR1. The results confirmed that high NPC1 expression is positively correlated with TGFBR1 levels in these tissues (Fig. 6d–f).

To explore whether NPC1 promotes HCC progression via TGFBR1 activation, we treated NPC1-overexpressing PLC/PRF/5 cells with LY2157299 (Galunisertib), a clinical inhibitor of TGFBR1 activation that has been investigated in clinical trials involving HCC patients[44–50]. Treatment with LY2157299 significantly attenuated the NPC1-induced enhancement of cell migration (Supplementary Fig. 8a, b and Fig. 6g, h). Additionally, in MHCC-97H cells with NPC1 knockdown and stable overexpression of TGFBR1 (Supplementary Fig. 8c), we observed that TGFBR1 overexpression rescued the metastatic potential of NPC1-knockdown cells in a mouse tail vein metastasis model (Fig. 6i–l).

Next, we explored the interaction between NPC1 and TGFBR1 by knocking down TGFBR1 in NPC1-overexpressing PLC/PRF/5 cells. This

knockdown significantly inhibited cell migration, though it did not affect cell proliferation or tumor growth in xenograft models (Supplementary Fig. 8d–i), indicating that NPC1-driven migration depends on TGFBR1. We also found that knocking down either NPC1 or TGFBR1 reduced migration in MHCC-97H cells (Supplementary Fig. 8k). In NPC1-knockdown cells, reintroducing wild-type NPC1 or the P691S mutant rescued the migratory phenotype, while reintroducing the NPC1 (Δ692-854) mutant failed to restore it (Supplementary Fig. 8j, k). These phenotypes were also validated in a mouse metastasis model, where knockdowns of both NPC1 and TGFBR1 reduced metastatic potential (Supplementary Fig. 8l, m), underscoring the critical roles of these proteins in HCC progression.

## Hepatic NPC1 deficiency suppresses TGF-β signaling and limits HCC progression

To explore NPC1's role in regulating TGF-β pathway and HCC progression in vivo, we utilized CRISPR-Cas9 system to generate liver-specific conditional *Npc1* knockout (*Cre^{Alb}Npc1^{F/F}*) and control mice (*Npc1^{F/F}*) (Fig. 7a) for analysis in the DEN-CCl₄ induced HCC model (Fig. 7b). This model is initiated by a single injection of the carcinogen diethylnitrosamine (DEN) followed by repeated administration of carbon tetrachloride (CCl₄). As expected, *Npc1* mRNA and protein levels were drastically reduced in the livers of *Cre^{Alb}Npc1^{F/F}* mice (Supplementary Fig. 9a, b). Remarkably, while control mice developed large liver tumors, *Cre^{Alb}Npc1^{F/F}* mice exhibited significantly smaller liver sizes and liver-to-body weight ratios at 24 weeks post-DEN injection (Fig. 7c, d and Supplementary Fig. 9c). These mice also showed fewer and smaller tumors (Fig. 7e, f and Supplementary Fig. 9d). Histological analysis demonstrated preserved reticulin structure in *Cre^{Alb}Npc1^{F/F}* tumors, in contrast to the disrupted reticulin patterns seen in control mice (Supplementary Fig. 9e). Liver injury markers-serum ALT, AST, and TBIL-were markedly reduced in *Cre^{Alb}Npc1^{F/F}* mice (Supplementary Fig. 9h–j), demonstrating ameliorated liver damage. Further IHC staining showed that *Cre^{Alb}Npc1^{F/F}* tumors had lower percentages of Ki67-positive cells (Supplementary Fig. 9f, g), indicating reduced tumor cell proliferation. Tumors also displayed diminished staining for EpCAM and cytokeratin 19 (Fig. 7g), both markers of poor prognosis and invasiveness in human HCC[51]. GRP78, a marker of aggressive disease[52], was similarly reduced in *Cre^{Alb}Npc1^{F/F}* mice (Fig. 7g). Importantly, *Npc1* knockout resulted in decreased expression of TGFBR1 protein and its downstream effector p-SMAD2 (Fig. 7h), reinforcing NPC1's role in regulating TGF-β signaling. Survival was significantly longer in *Cre^{Alb}Npc1^{F/F}* mice compared to controls (Fig. 7i).

We extended these observations to a MYC-driven HCC model by crossing *Cre^{Alb}Npc1^{F/F}* mice with *H11-CAG-LSL-Myc* mice to generate homozygous *Npc1* knockout mice (*Cre^{Alb}Npc1^{F/F}Myc*) (Fig. 7j). In this model, *Npc1* knockout significantly reduced tumor sizes and weights (Fig. 7k–m and Supplementary Fig. 9k). Consistent with the DEN-CCl₄

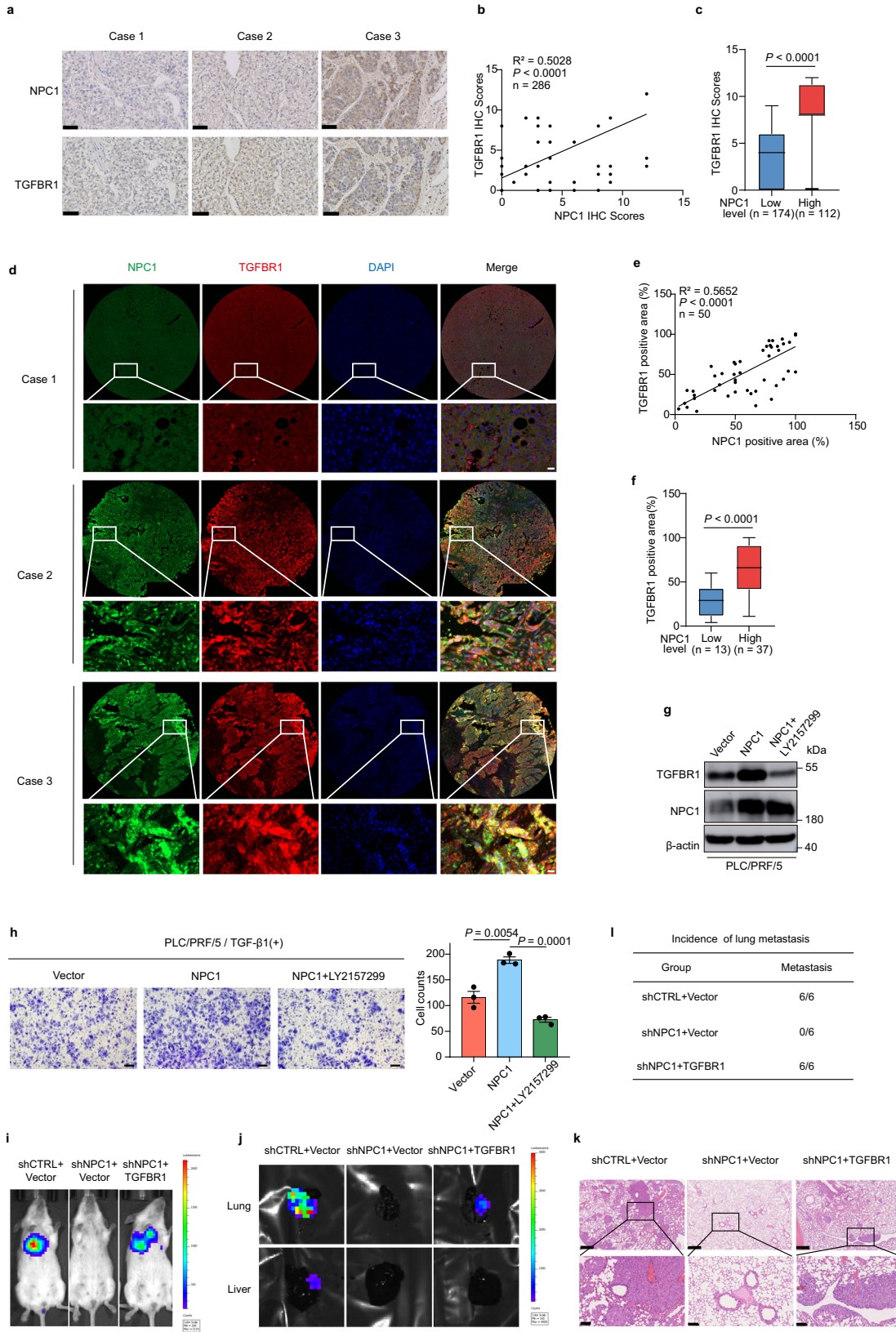

model, hepatic *Npc1* knockout inhibited TGF-β signaling and tumor progression in the MYC-driven HCC model (Fig.7n, o and Supplementary Fig. 9l–q).

Using an inducible *Npc1* knockout model with tamoxifen (TAM)-inducible Cre recombinase (ERT2-Cre) in DEN-CCl$_4$-induced HCC model, we confirmed that *Npc1* deficiency for an eight-week period

significantly reduced tumor burden, tumor number, and liver-to-body weight ratios (Fig. 7p–s and Supplementary Fig. 9r, s). Histological analysis further confirmed tumor regression in the *Npc1*-knockout mice (Fig. 7t). In addition, therapeutic targeting of NPC1 in MYC-driven HCC using AAV8-shNPC1 significantly reduced tumor size and number (Supplementary Fig. 9t–x), with knockdown efficiency validated by

**Fig. 6 | TGFBR1 is crucial for NPC1-mediated promotion of HCC progression.**
**a** Representative images from IHC staining of NPC1 and TGFBR1 in HCC tissues
($n = 286$ biologically independent samples); scale bars, 50 μm. **b** The Pearson correlation analysis between NPC1 level and TGFBR1 level in HCC tissues. **c** The analysis
of TGFBR1 IHC score in HCC tissues with low ($n = 174$ biologically independent
samples) or high ($n = 112$ biologically independent samples) NPC1 level.
**d** Representative images from Immunofluorescence staining of NPC1 and TGFBR1
in HCC tissues ($n = 10$ biologically independent samples); scale bars, 20 μm. **e** The
Pearson correlation analysis between NPC1 positive area and TGFBR1 positive area
in HCC tissues ($n = 50$ regions). These multiplexed IF staining were performed on
ten HCC tissue sections from HCC patients, qualifying an average of five regions per
sample. **f** The analysis of the percentages of TGFBR1 positive area in HCC tissues
with low ($n = 13$ regions) or high ($n = 37$ regions) NPC1 level. **g** Immunoblot analysis
of TGFBR1 and NPC1 expression in NPC1-overexpression PLC/PRF/5 cells with or
without TGFBR1 inhibitor LY2157299 (10 μM) treatment. **h** Transwell assay was

performed in cells related to (**g**); scale bars, 100 μm. **i-l** $1 \times 10^6$ Luciferase-expressing
HCC cells (MHCC-97H) were injected into NOD SCID mice by tail vein. The mice
were euthanized 8 weeks later by a cervical dislocation. Representative images of
whole body luminescence monitoring of NOD SCID mice injected via tail vein with
HCC cells 8 weeks after injection (**i**). Lung and liver tissues were isolated for analysis
of IVIS imaging (**j**). Representative H&E staining images of lung tissues are shown;
scale bars, 500 μm; insets: fivefold magnification; scale bars, 100 μm (**k**). The
incidence of lung metastasis in mice (**l**) ($n = 6$ mice per group). Data are presented
as the mean ± s.e.m. $n = 3$ (**h**) biologically independent samples. In the box plots, the
middle bar represents the median, and the box represents the interquartile range;
bars extend to 1.5× the interquartile range. Statistical significance was determined
by two-sided Pearson correlation test (**b, e**), two-sided Mann–Whitney U test (**c, f**)
or two-tailed unpaired Student's t-test (**h**). **g, h** Data were verified in three independent experiments. Source data are provided as a Source Data file.

## Discussion

NPC1 is a late endosomal/lysosome membrane protein involved in the
transport of low-density lipoprotein-derived cholesterol into cells and
virus entry, such as with Ebola[19,21]. Mutations in NPC1 cause Niemann-
Pick type C disease, a rare lipid storage disorder[12,53,54]. Despite its well-
characterized function in cholesterol metabolism, research into NPC1's
role in cancer is limited. Only a few reports suggest its upregulation in
some cancers, where it may contribute to proliferation and
invasion[7,55,56]. Here, we identify a pro-tumorigenic function for NPC1 in
hepatocellular carcinoma (HCC) using transgenic mice with *Npc1*
deletion in hepatocytes.

This study also has several implications for both TGF-β and NPC1
biology. TGF-β signaling regulates a variety of cellular processes,
including cell proliferation, differentiation, apoptosis, plasticity, and
migration[57]. Dysregulated TGF-β signaling has been linked to cancer
progression, stemness, and therapy resistance[45]. TGF-β signaling is
tightly regulated at different levels of the pathway. Regulation of TGF-β
signaling critically depends on the modulation of TGFBR1 activity and
its stability. The receptor's degradation is mediated not only by
caveolar-dependent endocytosis but also via clathrin-mediated internalization into early endosomes, followed by transport to late endosomes. The widely accepted model suggests that lipid rafts and
caveolae play a negative role in TGF-β signaling by facilitating the
turnover of TGF-β receptors, a process likely involving the recruitment
of SMAD7 and SMURF, which subsequently induce receptor
ubiquitination[37]. While literature has reported that TGFBR1 can localize
to caveolae and that NPC1 may colocalize with caveolin-1 under certain
conditions[37-40], our findings suggest that in HCC cells, the TGFBR1-
NPC1 complex is primarily localized within lysosomes. This suggests a
potential role for NPC1 in stabilizing TGFBR1 in lysosomes, rather than
promoting receptor turnover through lipid rafts/caveolae. Previous
studies indicate that SMAD7 can stably interact with TGFBR1 or
SMURF2 even in the absence of exogenous ligand stimulation, with
TGF-β potentially enhancing this interaction. Given that SMAD7 acts as
an inhibitor in the early stages of TGF-β signaling[58], our findings further
suggest that NPC1 inhibits the interaction between TGFBR1 and
SMAD7, thereby promoting TGFBR1 stability at a very early stage in the
TGF-β signaling pathway, even without TGF-β1 stimulation (Fig. 8). The
regulation we found occurs independent of ligand, consistent with
previous studies that TGFBR1 was constantly internalized and recycled
in the absence and presence of ligand[34-36]. We acknowledge that the
overexpression of the TGFBR1-mCherry-His construct may have influenced the observed localization patterns. This potential limitation
arises from the fact that overexpression could lead to artificial

redistribution of TGFBR1, which may not fully reflect its physiological
behavior. This should be carefully considered when interpreting the
data. To address this, further studies are needed to explore the role of
NPC1 in TGFBR1 degradation under more physiologically relevant
conditions, including using endogenous TGFBR1 levels and alternative
experimental approaches.

Structurally, NPC1 contains 13 transmembrane segments (TMs)
and three luminal domains (NTD, MLD, CTD), with TMs 3-7 forming the
sterol-sensing domain (SSD), which is conserved in proteins involved
in cholesterol metabolism. Previous studies on NPC1's structure show
that cholesterol first binds NPC2 and is then transferred to NPC1's NTD,
eventually entering a tunnel connected to SSD[59]. NPC1 is essential for
Ebola virus infection, that the domain C of NPC1 binds to the virus
glycoprotein (GP), independent of its known function in cholesterol
transport[18-21]. Interestingly, in our study, we found that the P691S
mutation in NPC1's SSD—a classical mutation that disrupts cholesterol
transport—did not affect its binding to TGFBR1. Reintroducing NPC1
(P691S) restored TGFBR1 levels and cell migration, highlighting a
cholesterol-independent function for NPC1 in cancer progression.
Further, our truncation experiments demonstrated that the 692-854
region of NPC1's TM domain is essential for its interaction with
TGFBR1, similar to NPC1's interaction with STING, which promotes
STING degradation and inhibits its signaling[60]. These interesting findings suggest that the interaction between NPC1 and other proteins may
play an important role in protein stability. These results underscore
that NPC1's interaction with other proteins may regulate their stability,
presenting an additional function beyond cholesterol transport.

Despite over a century of research on Niemann-Pick disease type
C and extensive structural studies on NPC1, its function in cancer has
been relatively unexplored. Our findings demonstrate that NPC1 promotes HCC progression by stabilizing TGFBR1 and facilitating tumor
cell migration. NPC1's influence on migration is likely related to its role
in membrane trafficking and integrin recycling, as reported in previous
studies[61,62]. These studies have linked NPC1 to cell migration via regulation of cholesterol levels and focal adhesion dynamics. However,
our study highlights an alternative, cholesterol-independent mechanism by which NPC1 interacts with TGFBR1 to promote cancer progression. Future research is needed to further elucidate the full scope
of NPC1's functions in cancer, particularly its interactions with other
signaling pathways. Regrettably, we did not use NPC1-specific inhibitors in this study to test its therapeutic effect on HCC. Although NPC1
SSD domain contains a U18666A-inhibitable site, while U18666A is a
widely used inhibitor of NPC1, it may also have broader effects on
cholesterol-related pathways[33,63]. To validate the therapeutic potential
of NPC1, we employed an inducible *Npc1* knockout model and AAV8-
mediated hepatic knockdown, both of which confirmed the tumor-
suppressive effects of NPC1 inhibition. Our findings reveal a
cholesterol-transport-independent role for NPC1 in HCC and suggest
therapeutic opportunities targeting NPC1 protein for HCC patients.

Western blotting (Supplementary Fig. 9y). These findings establish
NPC1 as a potential therapeutic target in HCC. Collectively, these data
provide strong evidence supporting the vital role of NPC1 in regulation
the TGF-β pathway and promoting HCC progression in vivo.

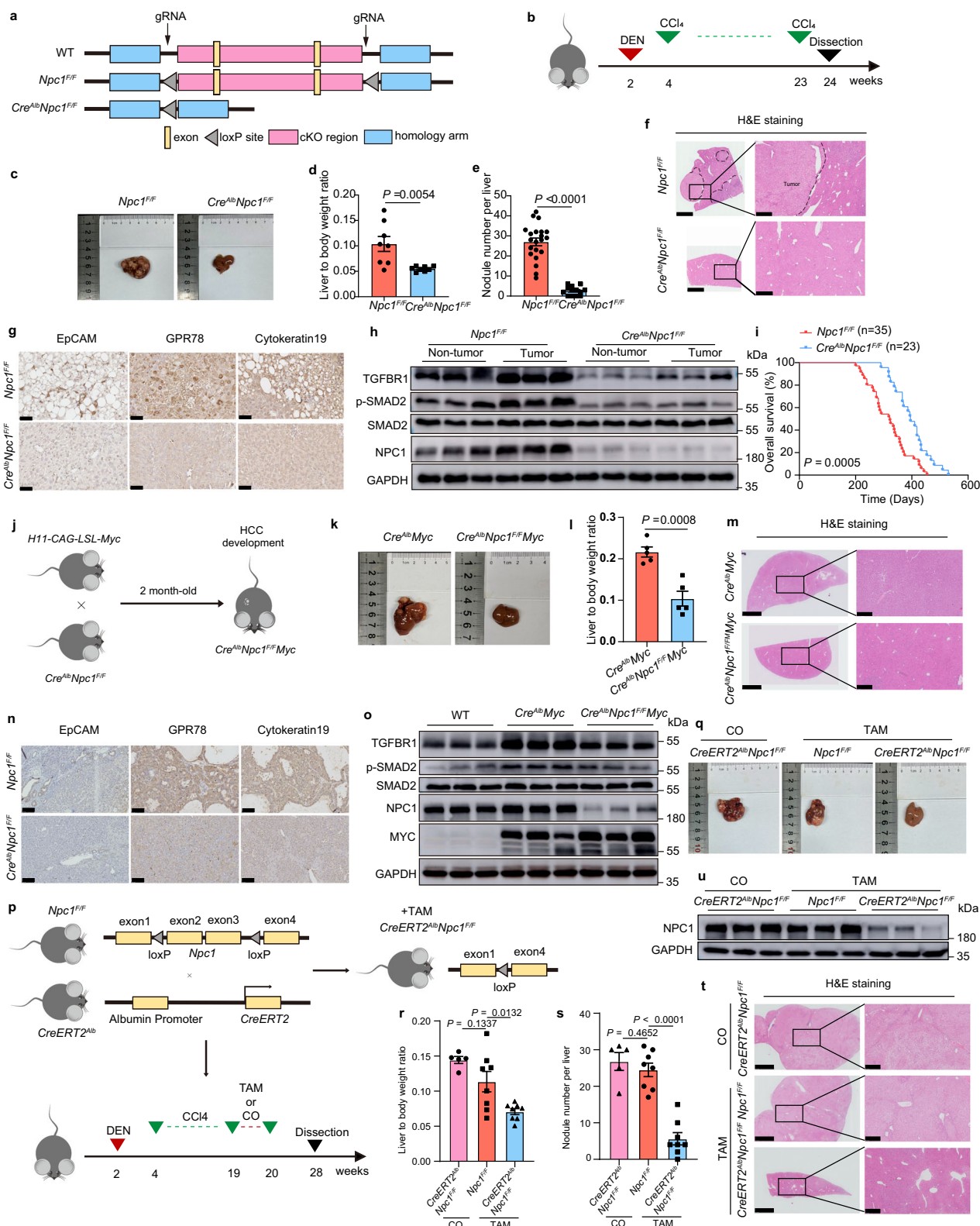

## Methods

### Ethics statement

The animal care and experimental protocols were approved by the Institutional Animal Care and Use Committee (IACUC) of the National Center for Protein Sciences, Beijing, China (NCPSB).

## Cell Lines

Human HCC cell lines (MHCC-97H, HepG2, PLC/RPF/5) and HEK-293T cells were used in this study. MHCC-97H cells were obtained from Liver Cancer Institute, Zhongshan Hospital, Fudan University. HepG2 (ATCC, HB-8065), HEK-293T (SCSP-502) and PLC/RPF/5 (TCHu119)

**Fig. 7 | Hepatic NPC1 deficiency inhibits HCC tumorigenesis and progression and TGF-β pathway. a** Construction of *Npc1*-conditional knockout (*Cre^{Alb}Npc1^{F/F}*) mice. **b**, *Npc1^{F/F}* and *Cre^{Alb}Npc1^{F/F}* mice were treated with DEN followed by twenty injections of CCl₄ to construct HCC mouse model. **c** Representative images of *Npc1^{F/F}* and *Cre^{Alb}Npc1^{F/F}* mouse livers with HCC. **d**, **e** liver to body weight ratio (*n* = 8 mice) (**d**) numbers of nodules per liver (*n* = 21 mice in *Npc1^{F/F}* and *n* = 13 mice in *Cre^{Alb}Npc1^{F/F}* groups) (**e**) of the indicated mice. **f**, **g** Representative H&E staining images (**f**), Representative IHC staining images of EpCAM, GPR78 and Cytokeratin19 (**g**) of the indicated mouse livers. H&E staining scale bars: left panels, 2.5 mm; right panels, 500 μm. IHC staining scale bars, 50 μm. **h** TGFBR1, p-SMAD2, SMAD2, and NPC1 protein expression in the indicated mouse non-tumor and tumor tissues. **i** Kaplan–Meier survival curves for *Npc1^{F/F}* (*n* = 35) or *Cre^{Alb}Npc1^{F/F}* (*n* = 23). **j** Scheme used to establish the model of spontaneous HCC with targeted *Myc* knock-in and *Npc1* knockout in the liver. **k** Representative images of *Cre^{Alb}Myc* and *Cre^{Alb}Npc1^{F/F}Myc* mouse livers with HCC. **l** liver to body weight ratio (*n* = 5) in the indicated mice.

**m**, **n** Representative H&E staining images (**m**), Representative IHC staining images of EpCAM, GPR78 and Cytokeratin19 (**n**) of the indicated mouse livers. H&E staining scale bars: left panels, 2.5 mm; right panels, 500 μm. IHC staining scale bars, 50 μm. **o** TGFBR1, p-SMAD2, SMAD2, and NPC1 protein expression in *WT*, *Cre^{Alb}Myc* and *Cre^{Alb}Npc1^{F/F}Myc* mouse liver tissues. **p** Schematic diagram of tamoxifen-induced liver-specific *Npc1* knockout mouse generation and the treatment plan in the DEN-CCl₄-induced HCC model. **q** Representative liver images of the indicated group. **r**, **s** liver to body weight ratio (**r**), numbers of nodules per liver (**s**) of *Npc1^{F/F}* mice treated with coil (CO) (*n* = 5), *CreERT2^{Alb}Npc1^{F/F}* mice treated with coil or tamoxifen (TAM) (*n* = 8). **t** Representative H&E staining images of the indicated mouse livers. Scale bars: left panels, 2.5 mm; right panels, 500 μm. **u** NPC1 protein expression in the indicated mouse tissues. Data are presented as means ± s.e.m. Statistical significance was determined by two-tailed unpaired Student's t-test (**d**, **e**, **l**, **r**, **s**) or log-rank test (**i**). These experiments (**f**, **g**, **h**, **m**–**o**, **t**, **u**) were repeated three times independently with similar results. Source data are provided as a Source Data file.

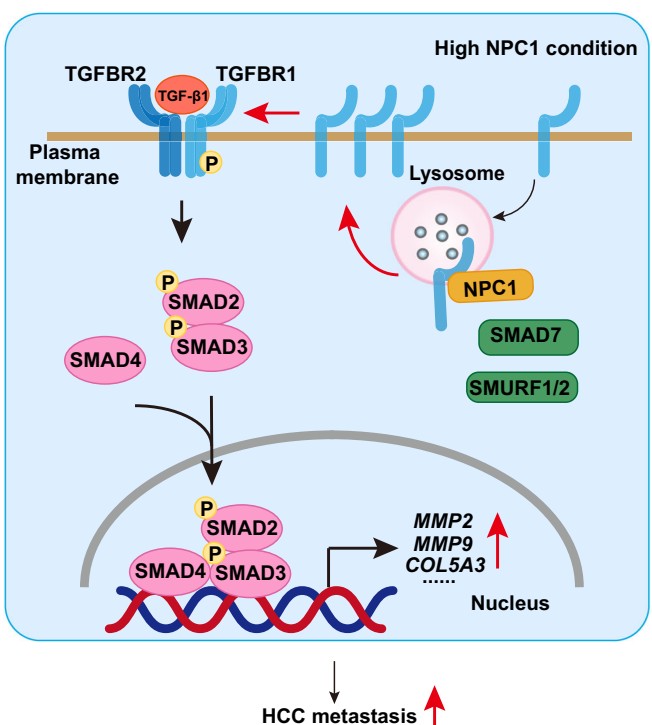

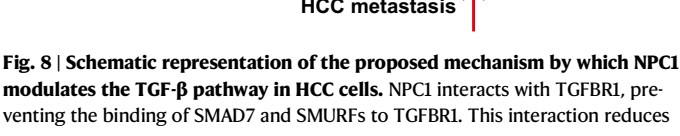

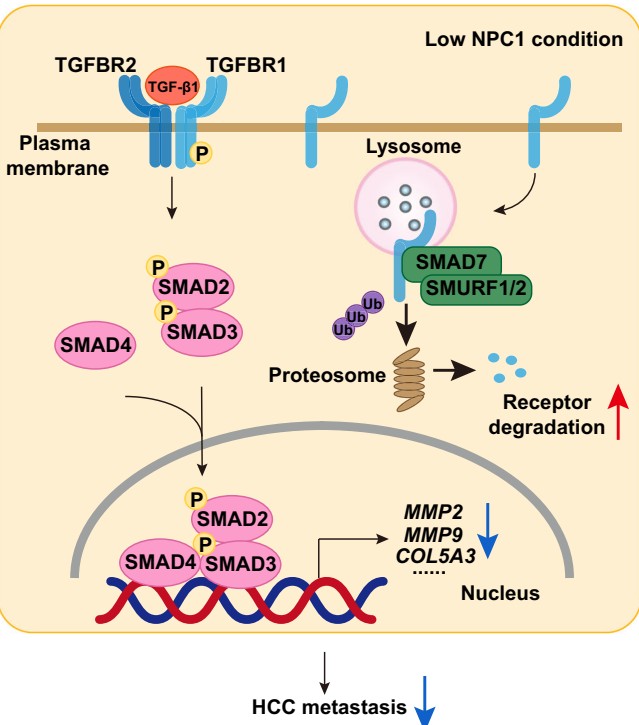

**Fig. 8 | Schematic representation of the proposed mechanism by which NPC1 modulates the TGF-β pathway in HCC cells.** NPC1 interacts with TGFBR1, preventing the binding of SMAD7 and SMURFs to TGFBR1. This interaction reduces TGFBR1 ubiquitylation and degradation, enhancing its stability and promoting TGF-β pathway activation in HCC cells.

cells purchased from Stem Cell Bank/Stem Cell Core Facility, SIBCB, CAS. Cells were maintained in DMEM (GIBCO) supplemented with 10% (v/v) fetal bovine serum (GIBCO) at 37°C in 5% CO₂. All the cell lines were authenticated by STR profiling and confirmed to be free of mycoplasma contamination.

## Mice

Mice were housed under specific pathogen-free (SPF) conditions in a temperature-controlled facility with a 12-hour light/dark cycle (08:00–20:00 light, 20:00–08:00 dark), and relative humidity maintained at 40-50%. Mice had free access to food and water throughout the study. All animals were maintained on normal chow diet (SPF-F02-003) until treatment. Age- and sex-matched mice were used for experiments. Liver-specific conditional *Npc1*-knockout mice (*Cre^{Alb}Npc1^{F/F}*, 2-4 months old, 3 males and 4 females) *Npc1^{F/F}* (2-4 months old, 8 males and 7 females) and *CreERT2^{Alb}* (6-8 weeks old, 2 males) in the C57BL/6 J background were made by Cyagen with

CRISPR-Cas9. Non-obese diabetic/severe combined immunodeficiency (NOD/SCID) mice (5 weeks old, female) were purchased from Beijing Vital River Laboratory Animal Technology. NCG mice (5 weeks old, male) were purchased from GemPharmatech Co. Ltd (Nanjing, China). *H11-CAG-LSL-Myc* mice (Cat. No. NM-KI-00039, 2-4 months old, 2 males and 2 females) were purchased from Shanghai Model Organisms.

## TMA and immunohistochemistry

Four TMA chips were purchased from Shanghai Outdo Biotech (LivH180Su08, HLivH180Su15, T16-855TMAB, TFHCC-02). Immunohistochemistry was performed as previously described[6]. Briefly, the tissues were exposed to primary antibodies overnight at 4°C, after which they were incubated with secondary antibodies. Immunostaining was performed according to the protocol provided by the Ventana automated staining system (Ventana Medical Systems). The intensity of NPC1 or TGFBR1 proteins was scored according to a previously

reported method[54], blindly and independently, by two trained pathologists. Each specimen was assigned a score based on both the intensity of staining in the nucleus, cytoplasm, and/or membrane, and the percentage of positively stained cells. The intensity was scored as follows: no staining = 0, weak staining = 1, moderate staining = 2, and strong staining = 3. The percentage of stained cells was rated as: 0% = 0, 1-25% = 1, 26-50% = 2, 51-75% = 3, and 76-100% = 4. The final immunoreactive score was calculated by multiplying the intensity score by the percentage score, resulting in a value ranging from 0 (minimum) to 12 (maximum). Image-Pro Plus software was used to quantify the number of Ki67-positive cells per field in tumor lesions. The antibodies used in the IHC assays are listed in Supplementary Data 2.

## Plasmids and established stable cells

Lentiviruses carrying shRNA targeting human NPC1 or SMAD7 were purchased from OBiO Technology (Shanghai) Co., Ltd. Lentiviruses carrying shRNA targeting human TGFBR1 were purchased from GenePharma (Shanghai) Co.,Ltd. The shRNA sequences are listed in Supplementary Data 3. HA-tagged NPC1, HA-tagged NPC1 (P691S) mutant, HA-tagged NPC1 deletion mutants, mCherry-His tagged TGFBR1, and EGFP-tagged SMAD7 overexpression lentiviral vectors and plasmids were custom-designed and synthesized by OBiO Technology (Shanghai) Co., Ltd. Information on the overexpression plasmids is provided in Supplementary Data 4.

## Small Interfering RNA (siRNA) and cell transduction

siRNAs targeting SMURF1 or SMURF2 were designed and synthesized by OBiO Technology (Shanghai) Co., Ltd. For cell transduction, Lipofectamine 2000 was used to perform the transfection according to the manufacturer's protocol. The sequences of the siRNAs are shown in Supplementary Data 3.

## Proliferation assay

Cells were plated into 96-well plates at a density of approximately 7,000 cells per well. CCK8 reagent was then added to each well at a final concentration of 10%, and the cells were incubated for 1 h. Absorbance was measured at 450 nm. Data analysis was performed using GraphPad Prism 8 software.

## Cell migration and invasion assays

For the transwell migration or invasion assays, cells were trypsinized and seeded onto the upper chamber of a 24-well migration (3422, Corning) or invasion (354480, Corning) plate. The lower chamber was filled with DMEM supplemented with 10% serum. After 24 h of incubation for the migration assay or 48 h for the invasion assay, the filters were removed, and the cells on the membrane were fixed with methanol. Migrated cells on the underside of the membrane were stained with 0.5% crystal violet. The excess dye was washed off with water, and the cells were then examined under a microscope.

## Western blotting

Protein lysates from cells were prepared using RIPA buffer (50 mM Tris-HCl, pH 8.0, 150 mM NaCl, 1 mM EDTA, 0.1% SDS and 1% NP-40) with 1% protease and phosphatase inhibitor cocktail (Thermo Fisher Scientific, 78441). The lysates were heated at 100 °C for 10 min and then subjected to electrophoresis on 6-12% SDS-polyacrylamide gels. Detailed information on the antibodies used can be found in Supplementary Data 2.

## Proteome profiling of HCC cell lines with NPC1 knockdown

HepG2 and PLC/PRF/5 cells were divided into two groups, each with three biological replicates: one group treated with scramble shRNA and the other with NPC1 shRNA. For protein extraction and trypsin digestion, cell samples were first lysed in buffer containing 1% sodium deoxycholate, 10 mM Tris (2-carboxyethyl) phosphine, 40 mM 2-

chloroacetamide, and 100 mM Tris-HCl (pH 8.8). The lysates were then heated at 95 °C for 10 min and subjected to sonication for 3 min (3 s on, 3 s off, at 30% amplitude). After centrifugation at 16,000 g for 10 minutes at 4 °C, the supernatants were collected. For digestion, 100 μg of protein (concentration measured using Thermo Nanodrop One) was treated overnight with trypsin (Promega, V528A) at 37 °C, and the reaction was stopped with 1% formic acid. Precipitated sodium deoxycholate was removed by centrifugation at 16,000 g for 10 min at 4 °C, and the supernatants were collected, desalted, vacuum-dried, and stored at −80 °C until further analysis.

For liquid chromatography-tandem mass spectrometry (LC-MS/MS) analysis, the dried peptides from shCTRL and shNPC1 HepG2 or PLC/PRF/5 cells were re-dissolved in Solvent A (0.1% formic acid in water) and loaded onto a homemade trap column (100 μm × 2 cm, particle size: 1.9 μm, pore size: 120 Å, SunChrom). The peptides were separated on a homemade 150 μm × 30 cm silica microcolumn (particle size: 1.9 μm, pore size: 120 Å, SunChrom) with a gradient of 5% to 35% mobile phase B (80% acetonitrile and 0.1% formic acid) at a flow rate of 600 nL/min over 140 minutes. After a 10-minute wash with 95% mobile phase B, the peptides were ionized at 2 kV. MS was conducted using data-dependent acquisition (DDA) mode. Raw MS files were analyzed using MaxQuant (version 1.6.1.0)[64] with default parameters against the human UniProt database (version 20180705). Further differencial analysis and functional enrichment analysis were performed using Perseus software[56] and the WebGestalt web tool[65].

## Quantitative RT-PCR

Total RNA was extracted from cells or tissues according to the manufacturer's instructions using TRIzol reagent (CWBIO, CW0580). Complementary DNA was synthesized from 1-3 μg of RNA with PrimeScript™ RT Master Mix (Takara, RR036A). qPCR was conducted using SYBR Green Master Mix (Vazyme, Q712-03), and primer sequences are provided in Supplementary Data 5. All samples were normalized to *Actb* for mouse samples or *ACTB* for human samples.

## Filipin III staining

A Cholesterol Cell-Based Detection Assay Kit (No. 10009779) from Cayman was used for the experiment. Filipin III staining was carried out following the manufacturer's instructions. Fluorescent staining was visualized using an upright fluorescence microscope (Nikon, Model No. 3000102).

## CHX chase assay

Cells were incubated with 50 μg/mL cycloheximide (CHX; Sigma, C7698) for the indicated times before being lysed in lysis buffer. Protein levels was determined by western blot analysis.

## Ubiquitylation Assay

TGFBR1-mCherry-His was transfected into HCC cells with stable NPC1 knockdown or overexpression. Cells were treated with MG132 for 8 h before collection. Cells were rinsed three times with prechilled phosphate-buffered saline (PBS) and subsequently lysed using immunoprecipitation buffer containing 6 M guanidine hydrochloride, 0.1 M NaH$_2$PO$_4$/Na$_2$HPO$_4$, 0.01 M Tris-HCl (pH 8.0), 5 mM imidazole and 10 mM β-mercaptoethanol. The lysates were then centrifuged at 14,000 g for 20 min at 4 °C. The resulting supernatants were incubated with Ni-NTA agarose beads at room temperature for 4 h. Beads were then washed respectively with washing buffer A (6 M guanidine hydrochloride, 0.1 M NaH$_2$PO$_4$/ Na$_2$HPO$_4$, 0.01 M Tris-HCl, pH 8.0, and 10 mM β-mercaptoethanol), washing buffer B (8 M urea, 0.1 M NaH$_2$PO$_4$/Na$_2$HPO$_4$, 0.01 M Tris-HCl, pH 8.0, and 10 mM β-mercaptoethanol), washing buffer C (8 M urea, 0.1 M NaH$_2$PO$_4$/Na$_2$HPO$_4$, 0.01 M Tris-HCl, pH 6.3, 10 mM β-mercaptoethanol and 0.2% Triton X-100) and washing buffer D (8 M urea, 0.1 M NaH$_2$PO$_4$/Na$_2$HPO$_4$, 0.01 M Tris-HCl, pH 6.3, 10 mM β-mercaptoethanol and 0.1% Triton X-100) for

5 min. Then the beads were incubated with buffer E (200 mM imidazole, 0.15 M Tris-HCl, pH 6.7, 30% glycerol, 0.72 M β-mercaptoethanol and 5% SDS) for 30 min at room temperature. The beads were subsequently boiled in loading buffer at 95 °C for 10 min. The resulting aliquots were collected and analyzed by western blot using antibodies against His, mCherry, K48-Ubiquitin, K63-Ubiquitin, or Ubiquitin.

## Immunoprecipitation

Cells were lysed in IP buffer (NP-40, Beyotime) supplemented with a protease inhibitor cocktail and incubated on ice for 2 h. The lysates were then centrifuged at 16,000 g for 10 min at 4 °C. Immunoprecipitation was performed using the specified primary antibody and Pierce™ protein A/G Magnetic Beads (Thermo Fisher Scientific, 88803), anti-DDDDK-tag mAb magnetic agarose (MBL, M185-10R), or anti-GFP mAb magnetic agarose (MBL, D153-10) at 4 °C. The immunocomplexes were thoroughly washed with the same buffer. Both lysates and immunoprecipitates were analyzed using the indicated primary antibodies, followed by detection with the corresponding secondary antibody and SuperSignal™ West Pico PLUS Chemiluminescent Substrate (Thermo Fisher Scientific, 34580). The primary antibodies used are listed in Supplementary Data 2.

## Immunofluorescence

Cells were fixed for 15 min with 4% paraformaldehyde at room temperature, washed twice with PBS and blocked in 10% normal goat serum for 1 h at room temperature. They were then incubated overnight with the primary antibody at 4°C. The antibodies are listed in Supplementary Data 2. The secondary antibodies, conjugated with fluorescein, were incubated for 1 h at room temperature. Nuclei were then stained with 4',6-diamidino-2-phenylindole (DAPI) for counterstaining. Confocal laser scanning was performed on a ZEISS LSM880 Confocal Microscope (Fig. 5c, Supplementary Fig. 6d, e, h) or a spinning disk confocal microscope (Supplementary Fig. 6b, c, g). The images in Supplementary Fig. 7a were collected using the Polar-SIM system (Airy Technology Co., Ltd., China). A 640 nm laser was used to excite mouse Alexa Fluor 633 with 2DSIM modality. The SIM reconstruction process was conducted using the Airy-SIM software with pre-processing (Dark) or post-processing (MRA). ImageJ software was applied to quantify colocalization using Mander's coefficient.

## Multiplexed Immunofluorescence (IF) Staining

Multiplex immunofluorescence staining was carried out using the PANO IHC Kit (Panovue, China) following the manufacturer's protocol. The stained slides were then scanned with the Digital Slide Scan system (AxioScan7), and individual scans for each slide were merged to generate a composite image. The resulting multilayer images were subsequently analyzed quantitatively using the ZEISS LSM880 Confocal Microscope.

## Animal studies

For subcutaneous xenograft experiments, NCG nude mice were randomly assigned to the designated groups and subcutaneously injected with the indicated cells, which stably expressed the specified shRNAs or constructs ($1 \times 10^7$ cells, mixed with 100 µl of Matrigel (ABW, 0827245) at a 1:1 ratio. Tumor progression was tracked, and tumor size was assessed using calipers. Tumor volume was determined using the following formula: (width$^2$ × length)/2.

For the NOD SCID or NCG mouse tail vein metastasis model, the mice were injected with the indicated luciferase-expressing cells through the lateral tail vein. All animals were sacrificed 8 weeks post-injection, and their lungs were surgically removed, fixed, and analyzed using hematoxylin and eosin (H&E) staining. All animal studies were approved by the Institutional Animal Care and Use Committee (IACUC) of the National Center for Protein Sciences, Beijing, China (NCPSB). The tumor size did not exceed 20 mm in any direction, in accordance

with the approval from our institutional review board. Mice were euthanized immediately if the tumor size exceeded 20 mm in any direction by the final measurement day. At the study endpoint, animals were euthanized through cervical dislocation while under anesthesia.

For the DEN/CCl$_4$-induced HCC model, 2-week-old male mice (*Npc1*$^{F/F}$ and *Cre*$^{Alb}$*Npc1*$^{F/F}$) were administered an intraperitoneal injection of DEN (50 mg/kg), followed by weekly CCl$_4$ injections (diluted 1:4 in corn oil) starting two weeks later at a dose of 2.5 ml/kg body weight, continuing for 20 weeks. Mice were euthanized 72 h after the final CCl$_4$ injection, and their livers were collected for biochemical, histological, and molecular analyses.

*CreERT2*$^{Alb}$*Npc1*$^{F/F}$ mice were generated by crossing *Npc1*$^{F/F}$ mice with *CreERT2*$^{Alb}$ mice. Two-week-old male mice (*Npc1*$^{F/F}$ and *CreERT2*$^{Alb}$*Npc1*$^{F/F}$) were injected intraperitoneally with DEN (50 mg/kg), followed by CCl$_4$ injections (diluted 1:4 in olive oil) at a dose of 2.5 ml/kg body weight once a week for 16 weeks. At 19 weeks of age, *CreERT2*$^{Alb}$*Npc1*$^{F/F}$ mice were randomly assigned to two groups and administered tamoxifen (TAM, Sigma-Aldrich, T5648, 75 mg/kg) or corn oil for 5 consecutive days via intraperitoneal injection and were sacrificed two months later. Tamoxifen was prepared by dissolving it in corn oil to a concentration of 20 mg/mL, and the solution was shaken overnight at 37 °C.

To establish the model of spontaneous HCC with targeted *Myc* knock-in and *Npc1* knockout in the liver, *Cre*$^{Alb}$*Npc1*$^{F/F}$ mice were crossbred with *H11-CAG-LSL-Myc* mice, resulting in homozygous *Npc1* knockout *Cre*$^{Alb}$*Npc1*$^{F/F}$ mice. Spontaneous liver cancer developed in 6- to 8-week-old mice following the mating of *H11-CAG-LSL-Myc* mice with *Cre*$^{Alb}$ mice. Eight-week-old *Cre*$^{Alb}$*Myc* and *Cre*$^{Alb}$*Npc1*$^{F/F}$ *Myc* mice were sacrificed, and their livers were used for biochemical, histological, and molecular analysis.

## Adeno-Associated Virus (AAV)-Mediated Gene Knockdown

An adeno-associated virus (AAV) delivery system was used to specifically knock down murine genes in mouse hepatocyte. The AAV cloning vectors AAV-shRNA and pAAV-TBG-GdGreen-miR30shRNA-WPRE were obtained from OBiO Technology (Shanghai) Co., Ltd. AAV8-shCTRL and AAV8-shNPC1 were generated by cloning a control shRNA (5'-GAAGTCGTGAGAAGTAGAA-3') or NPC1 shRNA (5'- CCCGTCTTACT-CAGTTACATA-3') into pAAV-TBG-GdGreen-miR30shRNA-WPRE. *Cre*$^{Alb}$*Myc* mice (4-week-old, male) were transduced with AAV serotype 8 vectors ($6 \times 10^{11}$ viral genome per mouse, via the tail vein) and euthanized at 8 weeks of age. Liver tumor tissues were collected for biochemical, histological, and molecular analysis.

## Histology

Livers and tumor tissues were fixed overnight in a neutral-buffered formalin solution, followed by dehydration and embedding in paraffin. The resulting sections were then used for Hematoxylin-Eosin and reticulin staining according to standard procedures.

## Biochemical assays

Blood samples were collected from the retro-orbital plexus of each mouse. The samples were then left at room temperature for 30 min to allow clotting. Afterward, the blood was centrifuged at 1,500 g for 10 min to separate the serum. Serum ALT, AST, TBIL levels were measured using an automatic chemical analyzer (Toshiba Biochemical Analyzer, model TBA 40FR).

## Statistics and reproducibility

Details regarding statistical analyses and sample sizes are provided in the figures, figure legends, and source data. While the data were presumed to follow a normal distribution, formal testing was not conducted. Animal experiments were randomly assigned to groups. Except for the analysis of IHC scores, data collection and analysis were not blinded to experimental conditions. Unless otherwise stated in the

figure legends, each experiment was conducted independently and yielded consistent results. Quantitative data are reported as mean ± s.e.m. Statistical analyses were performed in GraphPad Prism 8. Two-tailed unpaired Student's t-test, Mann-Whitney U test and two-way analysis of variance (ANOVA) were used to calculate $P$ values. Kaplan–Meier curves, with the log-rank test, were used to depict survival function from lifetime data. $P$ values are displayed in all figures, with values below 0.05 regarded as statistically significant.

### Reporting summary

Further information on research design is available in the Nature Portfolio Reporting Summary linked to this article.

## Data availability

The mass spectrometry proteomics data have been deposited with the ProteomeXchange Consortium (http://proteomecentral.proteomexchange.org) via the iProX partner repository[66,67] with the dataset identifier PXD046018 (https://www.iprox.cn//page/project.html?id=IPX0007268000). The remaining data are available within the Article, Supplementary Information or Source Data file. Source data are provided with this paper.

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

## Acknowledgements

This work was partially supported by the National Key R&D Program of China (No. 2021YFA1301601 to Y.J. and No. 2020YFE0202200 to Y.J.) and the National Natural Science Foundation of China (No. 82273243 to Y.J., No. 92168207 to Y.J., and No. 82090051 to C.L.). We thank Nieng Yan (Department of Molecular Biology, Princeton University) for providing the pCAG-NPC1-His-Flag plasmid. We thank the Imaging Facility (J. Chen), the Animal Facility (C. Qiu) and the Mass Spectrum Facility (S. Ji) of the National Center for Protein Sciences in Beijing (NCPSB) for their assistance.

## Author contributions

Y.J. and F.H. conceived the study and supervised experiments. S.L. designed experiments. S.L., L.Y., C.L., L.L., F.C. and X.Y. performed experiments. S.L. and L.Y. analyzed the data. S.L., Y.J., F.H., and L.Y. wrote the paper. All authors read and approved the manuscript.

## Competing interests

The authors declare no competing interests.
