## [Transparent Peer Review file · Nature Communications]

NPC1 controls TGFBR1 stability in a cholesterol transport-independent manner and promotes hepatocellular carcinoma progression

Corresponding Author: Professor Ying Jiang

Version 0:

Reviewer comments:

Reviewer #1

(Remarks to the Author)

In this paper, Li et al. found that NPC1 expression is up-regulated in Hepatocellular Carcinoma (HCC), correlating with poor prognosis. They then used stable overexpression and knockdown HCC cellular models to show that NPC1 expression impacts proliferation and migration of different HCC cell lines. These findings confirm results from a study that came out very recently (PMID: 38042211; this study should be cited). Subsequent proteomics analysis allowed them to reveal a link between NPC1 expression and the TGF- β pathway; NPC1 influences TGF- β signaling in a cholesterol-independent manner. Mechanistically, the authors find an intriguing interaction between NPC1 and TGF- β receptor 1 (TGFBR1) that seems to abrogate its ubiquitylation mediated by SMAD7/SUMRFs.

Finally, in vivo mouse models including generation of liver-specific Npc1cKO mice are used to confirm a role for NPC1 in TGF- β signaling-mediated oncogenicity in HCC.

Although their findings are interesting, mechanistically speaking is not clear how NPC1, which resides in endolysosomes, may be acting to influence TGF- β signaling. Previous work has shown that TGF- β and its cognate receptors traffic through the endosomal pathway (PMID: 19050695) and that TGF- β signaling is regulated by endocytosis. The authors should discuss more on this topic and in relation to what NPC1 may be doing in this context according to their data.

Altogether, I would recommend this work for publication if the authors address the following comments, which will further support their findings:

1) In lines 231-232, the authors mention that "NPC1 and TGFBR1 were colocalized in the cytoplasm of HCC cells". NPC1 is an endolysosomal membrane protein therefore, the authors need to show clear co-localization with TGFBR1 in this compartment, under both NPC1 over-expression and knockdown conditions. Images in Fig. 5c and Extended Fig.5b are not convincing. Does stimulation of cells with TGF- β redistribute TGFBR1 from the PM to lysosomes? co-localization should also be quantified. These data will further support their model.

2) Is sustained TGF- β signaling observed under NPC1 over-expression conditions? Previous work showed a role for endolysosomal ESCRT in TGF- β receptor down-regulation (PMCID: PMC7615189). Can authors provide dynamics of TGF- β signaling (time course experiments) under NPC1 over-expression?

3) Importantly, the authors should indicate relative mobility of molecular weight markers (in kDa) in ALL their immunoblots.

4) The authors use M β CD to investigate the role of cholesterol in relation to the TGF- β pathway. They should rather use U18666A which specifically inhibits NPC1-mediated cholesterol transport. Indeed, the authors should remove the claim (in Line 394) that U18666A is not a specific inhibitor of NPC1 and cite PMID: 26646182.

5) Degradation of TGFBR1 is mediated by both proteasomal and lysosomal pathways; while the authors show blots of the effect of MG132 (proteasomal inhibitor) and NH₄Cl (lysosomal inhibitor), quantitation (from different independent experiments) of these immunoblots should be plotted.

Minor comments:

- 1) Immunoprecipitations do not imply direct binding; therefore the authors should remove the claim that NPC1 and TGFBR1 directly interact.
- 2) Authors should cite relevant work in relation to the role of NPC1 in cell migration:
PMID: 24746815, PMID: 35022465.
- 3) Results shown in Fig 3 g should be better explained in the manuscript.
- 4) In lines 304-307 the authors comment on serum lipid/lipoprotein content of Npc1 cKO : "Furthermore, serum levels of triglyceride (TG), cholesterol (CHOL), high-density lipoprotein cholesterol (HDL-C), low-density lipoprotein cholesterol (LDL-C) and very low-density lipoprotein (VLDL) were all found to be lower in Npc1cKO mice compared to control mice". The authors should better explain these findings in relation to the NPC1-cholesterol-SREBP axis, which regulates transcription of multiple genes related to lipid metabolism and LDLR trafficking.
- 5) Please check for English grammar and typos throughout the text: For example, Line 150 is missing a word in this sentence "that the critical of NPC1" (should it read "critical role"?).

Reviewer #2

(Remarks to the Author)

In this study, Li et al. decipher the role of NPC1, which was first identified as a potential risk marker in HCC in their previous paper using proteomics (Jiang et al., 2019). But, one recent study also has suggested the oncogenic role and prognostic value of NPC1 in HCC via regulation of autophagy (Xu et al., 2024). Here, by using in vitro and in vivo models of HCC, the authors propose that NPC1 directly interacts with TGFBR1 and prevents the binding between TGFBR1 and the SMAD7/SMURFs complex to maintain the stability of TGFBR1, thus activating TGF- β signaling in a cholesterol transport-independent manner and promoting HCC progression. This is an interesting topic, and the novelty relies on the effect and mechanism of NPC1 on cholesterol-independent regulation. Although the authors provide a large amount of experimental data with good quality, the downstream mechanism via TGF- β signaling is well recognized in HCC, and the interaction between TGF- β and SMAD7/SMURFs is also well known, which dampens the novelty of their study. Overall, the following comments should be addressed to strengthen the overall scientific soundness of the manuscript:

1. In Figure 2 and extended data Fig.2, authors re-induced NPC1 in HCC cells to prevent off-target effects from shRNA knockdown. The NPC1 protein level in shNPC1+NPC1 group was much higher than that in the shCTRL group, whereas most of the function studies, including cell proliferation, migration, invasion, as well as the tail vein metastasis model, showed no difference between the two groups or only had a partial rescue effect. How do the authors explain these findings? According to Figures 2a-e, it is anticipated that re-induced NPC1 should promote metastasis compared to Vector.
2. Besides the metastasis model, Subcutaneous/orthotopic tumor inoculation should be conducted to verify whether NPC1 knockdown and reinduction could promote tumor growth.
3. In Figures 3a-f, please clarify the statement where TGF- β signaling downstream proteins p-SMAD2 and p-SMAD3 in NPC1-overexpressed HCC cells seemed to be only upregulated and detectable under TGF- β 1 treatment. Authors also claimed that NPC1 knockdown decreased HCC cell migration capacity independent of TGF- β 1 treatment. Does this mean that despite TGFBR1 increase, SMAD2/3 activation is not the crucial signaling regulated by NPC1 in driving HCC metastasis?
4. In the Method of cell invasion assays, NPC1 expression did influence HCC cell proliferation (extended Data Fig.2a-d) and showed a significant difference at 48 or 72 hours. Therefore, 72 hours selected for invasion assays is not convincing, and a shorter incubation period should be chosen to exclude the effects on cell proliferation.
5. Have the authors detected the phospho-TGFBR1 (activated state) alteration by NPC1? If so, through what kind of mechanism? Does TGFBR1 mainly rely on phosphorylation to exert its effect on SMAD2/3? Does NPC1 only increase pan-TGFBR1, which is then phosphorylated by TGFBR2? Additionally, in Figure 6a, it is confusing that the inhibitor of TGFBR1 activation Galunisertib also inhibited pan-TGFBR1 expression level.
6. Does SMAD7/SMURFs share the same binding domain with SAMD2/3 to TGFBR1, and which one has a higher intrinsic binding affinity to TGFBR1? Besides its function on enhancing the degradation of TGFBR1, SMAD7/SMURFs might competitively impede the TGFBR1-SMAD2/3 binding. By inhibiting NPC1, TGFBR1's binding domain might be more accessible to SAMD2/3. These are some questions the author should carefully demonstrate.
7. In the DEN/CCI4-driven mouse HCC model, the authors demonstrated that Npc1cKO mice displayed fewer tumor lesions with lower serum levels of components involved in cholesterol transport, compared to control mice. However, the mechanism of NPC1-TGFBR1 axis-induced HCC metastasis illustrated is cholesterol-transport independent. Since such

model induces fibrosis and inflammation-associated HCC linked to lipid dysfunction, an alternative HCC mouse model of combined hydrodynamic injection of proto-oncogenes (e.g. c-myc) and/or tumor-suppressive genes (e.g. sgp53 and keap1) may be more suitable for in vivo function validation in this context.

8. The effect of NPC1 mainly depends on TGFBR1 expression. So does it mean that if an HCC patient with tumor cells lacked TGFBR1, NPC1 would not contribute to tumor progression? The authors should first validate this hypothesis by overexpressing NPC1 in TGFBR1 knockout HCC cells and compare their function to TGFBR1-WT cells. Then, in their clinical cohorts, the authors should demonstrate whether NPC1's prognostic value is only present in TGFBR1-co-expressing HCC patients, but not in TGFBR1-low and negative patients. Also, in their spontaneous HCC model, each mouse has a different expression of TGFBR1 on tumor cells. Thus, it is important to stratify the mouse tumor samples into TGFBR1-high and TGFBR1-low groups to evaluate the tumor size/nodule number/IHC/WB, as shown in Figures 7e-o.

9. Animal studies: please confirm if any subcutaneous mouse model was used as described in Methods. In addition, the authors claimed in the Discussion section that there has been no specific NPC1 inhibitor to test its therapeutic efficacy on HCC in mice until now. AAV8 therapy specifically targeting hepatic NPC1 in an orthotopic mouse model should be performed to validate its therapeutic potential. More importantly, the authors should compare the effects of NPC1 inhibition and TGFBR1 inhibition in vitro and in vivo, including migration/invasion assays, subcutaneous/orthotopic tumor inoculation, and tail-vein lung metastasis model.

10. Careful grammar check has to be carried out since the manuscript has multiple grammatical errors.

Reviewer #3

(Remarks to the Author)

The authors argue that NPC1 controls TGFbetaRI (TbRI) stability in a cholesterol-independent manner is promoting hepatocellular carcinoma progression. They have used conventional methods to knock down NPC1 in vitro and in vivo and show that NPC1 promotes tumor progression in vitro and in vivo. They propose that NPC1 is causing increased stability of TbRI by preventing Smad7 to recruit SMURFs, which targets TbRI for degradation.

The presented data are interesting and novel, but I have some concerns about their data and the conclusions they draw.

1, On line 101 the authors claim that NPC1 "directly interacts with" TbRI. To claim this, they should show direct interactions between the two proteins.

2, If NPC1 recruits Smad7 and SMURFs to the TbRI, they should show that this is dependent on TGFbeta treatment of cells. From the literature we know that "Smad7 recruits the HECT type of E3 ubiquitin ligases, Smurf1 and Smurf2. It binds to Smurfs in the nucleus and translocates into the cytoplasm in response to TGF- β and recruits the ubiquitin ligases to the activated type I receptor ALK5/TbRI, leading to the degradation of the receptor through the proteasomal pathway." Therefore, it would be of scientific interest to know in which subcellular localization the TbRI-NPC1 complex is located. The complex is shown by immunofluorescence in Figure 5c.

Concerns/Questions:

A, Is the TbRI-NPC1 complex localized in caveola?

B, would a mutant NPC1 lacking 692-854 region, not recruit TbRI to this subcellular localization?

C, what happens with this subcellular complex (TbRI-NPC1) in presence of TGFbeta stimulation of cells? Will treatment with Galunisertib change the co-localisation of TbRI and NPC1?

D, If NPC1 control TbRI protein stability, it would be interesting to know if NPC1 and TbRI are co-expressed in HCC tissue sections. This can be investigated with immunohistochemistry.

Does high expression of TbRI also correlate with poor prognosis for patients with HCC?

E, Discussion line 346; "Galunisertib is the only TGFbeta pathway inhibitor currently under clinical investigation in HCC patients" this should be revised; as far as I know is the effect of Galunisertib no longer investigated in clinical trials.

Version 1:

Reviewer comments:

Reviewer #1

(Remarks to the Author)

In this revised version of their manuscript, Li and colleagues have addressed basically all comments that I had previously raised. Moreover, they have also addressed point by point other reviewer's important concerns.

In spite of this, I believe that over-expression of TGFBR1 probably leads to over-interpretation of some of their conclusions regarding trafficking of this receptor. As the authors note, previous studies have shown TGFBR1 internalization independent of TGF β 1 stimulation, however, the recycling compartment has nothing to do with lysosomes. Here, in their rebuttal letter, the authors show lysosomal localization of TGFBR1 regardless of TGF β 1-mediated stimulation. This is somewhat unexpected, since I would expect an increase in the amount of receptor molecules that reach the lysosomal compartment (after internalization), where NPC1 resides. The fact that over-expressed TGFBR1 localizes in lysosomes at steady state (in

unstimulated conditions) makes me wonder whether this could be an artefactual effect of over-expression. If a good anti-TGFBR1 antibody exists for immunofluorescence experiments, labeling of endogenous TGFBR1 in both unstimulated and TGFβ1-stimulated cells would solve this issue. Otherwise, the authors should consider this potential pitfall in their over-expression system and clearly reflect this issue in the manuscript.

Finally, in their Revised Supplementary Fig. 4e, I noted that TGFBR1 protein levels increase from 4 to 24 h of TGFβ1 stimulation. Is this expected? Others have shown lysosomal-mediated TGFBR1 downregulation after TGFβ1 stimulation (PMID: 12717440 ; PMID: 30428352). Please provide some comment to this.

Altogether, I acknowledge the author's efforts to address the multiple concerns raised by myself and the rest of reviewers. Overall, the manuscript has increased its quality. Publication is recommended after addressing these two minor comments stated above.

Reviewer #2

(Remarks to the Author)

The authors have satisfactorily addressed my comments.

Reviewer #3

(Remarks to the Author)

The revised manuscript have included data from additional experiments as suggested. I am satisfied with the new data as presented in the revised manuscript and have no further questions.

Dear Editor,

Thank you for giving us the opportunity to respond to the reviewers' comments point by point, on our manuscript (ID: NCOMMS-23-61307A) entitled “**NPC1 controls TGFBR1 stability in a cholesterol transport-independent manner and promotes hepatocellular carcinoma progression**” for consideration for publication in *Nature Communications*. We appreciate very much the invaluable comments and suggestions from reviewers and editors. Thus, heartfelt thanks to you and the reviewers for the great effort and long-time put into the review of the manuscript. I promise, each comment or suggestion has been carefully considered and responded point by point. Responses to the reviewers and changes in the revised manuscript are described as follows.

Reviewer #1

In this paper, Li et al. found that NPC1 expression is up-regulated in Hepatocellular Carcinoma (HCC), correlating with poor prognosis. They then used stable overexpression and knockdown HCC cellular models to show that NPC1 expression impacts proliferation and migration of different HCC cell lines. These findings confirm results from a study that came out very recently (PMID: 38042211; this study should be cited). Subsequent proteomics analysis allowed them to reveal a link between NPC1 expression and the TGF- β pathway; NPC1 influences TGF- β signaling in a cholesterol-independent manner. Mechanistically, the authors find an intriguing interaction between NPC1 and TGF- β receptor 1 (TGFBR1) that seems to abrogate its ubiquitylation mediated by SMAD7/SMURFs.

Finally, in vivo mouse models including generation of liver-specific *Npc1* cKO mice are used to confirm a role for NPC1 in TGF- β signaling-mediated oncogenicity in HCC.

Although their findings are interesting, mechanistically speaking is not clear how NPC1, which resides in endolysosomes, may be acting to influence TGF- β signaling. Previous work has shown that TGF- β and its cognate receptors traffic through the endosomal pathway (PMID: 19050695) and that TGF- β signaling is regulated by endocytosis. The authors should discuss more on this topic and in relation to what NPC1 may be doing in this context according to their data.

Altogether, I would recommend this work for publication if the authors address the following comments, which will further support their findings:

Major comments:

Q1: In lines 231-232, the authors mention that “NPC1 and TGFBR1 were colocalized in the cytoplasm of HCC cells”. NPC1 is an endolysosomal membrane protein therefore, the authors need to show clear co-localization with TGFBR1 in

this compartment, under both NPC1 over-expression and knockdown conditions. Images in Fig. 5c and Extended Fig. 5b are not convincing. Does stimulation of cells with TGF- β redistribute TGFBR1 from the PM to lysosomes? co-localization should also be quantified. These data will further support their model.

A1: Thank you for your insightful suggestion. We completely agree on the need to clearly demonstrate the co-localization of NPC1 with TGFBR1. In response, we performed additional experiments to assess the subcellular localization of the NPC1-TGFBR1 complex.

To gain insights into the subcellular localization of the TGFBR1-NPC1 complex, we examined their co-localization patterns. Although prior studies reported that TGFBR1 undergoes sustained internalization and localizes to various cytoplasmic vesicles, including compartments marked by LAMP1 or caveolin-1¹⁻⁴, our results revealed that the TGFBR1-NPC1 complex predominantly colocalizes with the lysosomal marker LAMP1 rather than with caveolin-1 (Below Panel, Revised Fig. 5c and Revised Supplementary Fig. 5b-f). This finding indicates that, within HCC cells, the TGFBR1-NPC1 complex resides primarily in lysosomes.

Fig. 5c

Supplementary Fig. 5b

Supplementary Fig. 5c

Supplementary Fig. 5d

Supplementary Fig. 5e

Supplementary Fig. 5f

Fig. 5c, PLC/PRF/5 cells stably overexpressing TGFBR1-mCherry-His and NPC1-HA were immunostained with antibodies against HA and LAMP1 to determine the colocalization among TGFBR1, NPC1 and LAMP1 in PLC/PRF/5 cells. Representative images from three independent experiments are shown; scale bars, 10 μ m.

Supplementary Fig. 5b, HepG2 cells stably expressing TGFBR1-mCherry-His were immunostained with antibodies against NPC1 and LAMP1 to determine the colocalization among TGFBR1, NPC1 and LAMP1 in HepG2 cells.

Supplementary Fig. 5c, HepG2 cells stably expressing TGFBR1-mCherry-His and Caveolin-1 (CAV1)-Myc were immunostained with antibodies against Myc and LAMP1 to determine the colocalization among TGFBR1, CAV1 and LAMP1 in HepG2 cells.

Supplementary Fig. 5d-e, HepG2 cells stably expressing TGFBR1-mCherry-His were immunostained with antibodies against RAB5(d)/RAB11(e) and LAMP1 to determine the colocalization among TGFBR1, RAB5/ RAB11 and LAMP1 in HepG2 cells.

Supplementary Fig. 5f, Quantification of the colocalization between these four proteins (LAMP1, CAV1, RAB5 and RAB11) and TGFBR1 using the Mander's coefficient.

Representative images from three independent experiments are shown; scale bars, 10 μ m. Data are presented as the mean \pm s.e.m. Statistical significance was determined by two-tailed unpaired Student's t-test.

Notably, neither overexpression nor knockdown of NPC1 altered the lysosomal localization of TGFBR1 (Below Panel, Revised Fig. 5c and Revised Supplementary Fig. 5h, j, b, g, i).

Fig. 5c, PLC/PRF/5 cells stably overexpressing TGFBR1-mCherry-His and NPC1-HA were immunostained with antibodies against HA and LAMP1 to determine the colocalization among TGFBR1, NPC1 and LAMP1 in PLC/PRF/5 cells.

Supplementary Fig. 5h, PLC/PRF/5 cells stably overexpressing TGFBR1-mCherry-His and HA-tag were immunostained with antibodies against NPC1 and LAMP1 to determine the colocalization among TGFBR1, NPC1 and LAMP1 in PLC/PRF/5 cells.

Supplementary Fig. 5j, Quantification of the colocalization between TGFBR1 and LAMP1

using the Mander's coefficient in PLC/PRF/5 cells with NPC1 stable overexpression.

Supplementary Fig. 5b, HepG2 cells stably expressing TGFBR1-mCherry-His were immunostained with antibodies against NPC1 and LAMP1 to determine the colocalization among TGFBR1, NPC1 and LAMP1 in HepG2 cells.

Supplementary Fig. 5g, NPC1-knockdown HepG2 cells stably expressing TGFBR1-mCherry-His were immunostained with antibodies against NPC1 and LAMP1 to determine the colocalization among TGFBR1, NPC1 and LAMP1 in HepG2 cells.

Supplementary Fig. 5i, Quantification of the colocalization between TGFBR1 and LAMP1 using the Mander's coefficient in HepG2 cells with NPC1 stable knockdown

Representative images from three independent experiments are shown; scale bars, 10 μ m. Data are presented as the mean \pm s.e.m. Statistical significance was determined by two-tailed unpaired Student's t-test.

To further explore the dynamics of TGFBR1 localization, we examined PLC/PRF/5 (TGFBR1-mCherry-His) cells both with and without TGF- β 1 stimulation. We used anti-NPC1 and anti-LAMP1 antibodies to quantify the co-localization between TGFBR1 and LAMP1 under these conditions. Our data show that TGFBR1-NPC1 remains localized to lysosomes regardless of TGF- β 1 stimulation, with no significant difference in the co-localization coefficient between stimulated and unstimulated groups (Below Panel A, B). Previous studies indicate that SMAD7 can stably interact with TGFBR1 or SMURF2 even in the absence of exogenous ligand stimulation, with TGF- β potentially enhancing this interaction. Given that SMAD7 acts as an inhibitor in the early stages of TGF- β signaling⁵, our findings further suggest that NPC1 inhibits the interaction between TGFBR1 and SMAD7, thereby promoting TGFBR1 stability at a very early stage in the TGF- β signaling pathway, even without TGF- β 1 stimulation. The regulation we found occurs independent of ligand, consistent with previous studies that TGFBR1 was constantly internalized and recycled in the absence and presence of ligand⁶⁻⁸.

Legend: (A) PLC/PRF/5 cells stably expressing TGFBR1-mCherry-His with or without TGF- β 1 (10 ng/mL) treatment for 30 min were immunostained with antibodies against LAMP1 and NPC1 and then imaged by Structure Illumination Microscopy to determine the colocalization among TGFBR1, LAMP1 and NPC1 in indicated cells. The images were collected through the

Polar-SIM system (Airy Technology Co., Ltd., China). A 488 nm (or 640) laser were used to excite rabbit Alexa Fluor 488 (or mouse Alexa Fluor 633) with 2DSIM modality. The SIM reconstruction process was conducted using the Airy-SIM software with pre-processing (Dark) or post processing (MRA); Scale bar, 10 μ m. **(B)** Quantification of the colocalization between TGFBR1 and LAMP1 using the Mander's coefficient. Representative images from three independent experiments are shown; scale bars, 10 μ m. Data are presented as the mean \pm s.e.m. Statistical significance was determined by two-tailed unpaired Student's t-test.

Q2: Is sustained TGF- β signaling observed under NPC1 over-expression conditions? Previous work showed a role for endolysosomal ESCRT in TGF- β receptor down-regulation (PMCID: PMC7615189). Can authors provide dynamics of TGF- β signaling (time course experiments) under NPC1 over-expression?

A2: Thank you for your important question. To investigate the impact of NPC1 on TGF- β signaling dynamics, we conducted time course experiments in PLC/PRF/5 cells following the experimental design outlined in the paper you mentioned⁸. Under conditions of NPC1 overexpression, we observed sustained SMAD2 phosphorylation over a 24-hour period, whereas the control cells showed attenuation of the signal (Below Panel, Revised Supplementary Fig. 4e). These findings suggest that NPC1 overexpression leads to prolonged TGF- β signaling, likely due to impaired receptor down-regulation.

Supplementary Fig. 4e

Supplementary Fig. 4e, PLC/PRF/5 cells with stable overexpression of NPC1 were stimulated with TGF- β 1 (10 ng/mL) for the times indicated. Levels of p-SMAD2, SMAD2, TGFBR1 and NPC1 were assayed by western blot. Data are presented as the mean \pm s.e.m. of three independent experiments. Statistical significance was determined by two-tailed unpaired Student's t-test.

Q3: Importantly, the authors should indicate relative mobility of molecular weight markers (in kDa) in ALL their immunoblots.

A3: Thank you for your suggestion. We have revised our immunoblots to include the relative mobility of molecular weight markers (in kDa) in all figures, as you recommended. We appreciate your attention to detail and believe this change will enhance the clarity and accuracy of our data presentation.

Q4: The authors use M β CD to investigate the role of cholesterol in relation to the

TGF- β pathway. They should rather use U18666A which specifically inhibits NPC1-mediated cholesterol transport. Indeed, the authors should remove the claim (in Line 394) that U18666A is not a specific inhibitor of NPC1 and cite PMID: 26646182.

A4: Thank you for your valuable suggestion. As per your recommendation, we treated cells with U18666A at various concentrations to assess its effects on the TGF- β pathway. Our results revealed that U18666A treatment did not alter TGFBR1 levels or downstream p-SMAD2 levels, consistent with our findings using M β CD (Below Panel, Revised Supplementary Fig. 3i-l).

We acknowledge the reference you provided, which identifies NPC1 as a primary target of U18666A⁹. However, our original statement in the manuscript was based on reports indicating that U18666A can inhibit additional proteins involved in cholesterol biosynthesis, as noted in earlier literature¹⁰. We have updated the manuscript to clarify that while U18666A is a widely used inhibitor of NPC1, it may also have broader effects on cholesterol-related pathways.

Our findings with both U18666A and M β CD suggest that NPC1's regulation of the TGF- β pathway occurs independently of its cholesterol transport function, supporting our conclusion that cholesterol modulation does not significantly impact TGF- β signaling under these conditions.

Supplementary Fig. 3i-l, HepG2 cells were treated with M β CD (**i**) or U18666A (**k**), then analyzed by western blot to measure whole cell expression of TGFBR1 or the activity of TGF- β pathway. Cells related to (**i**, **k**) were fixed and stained with filipin to label free cholesterol accumulated in LE/Ly. Representative images from three independent experiments are shown; scale bars, 10 μ m.

Q5: Degradation of TGFBR1 is mediated by both proteasomal and lysosomal pathways; while the authors show blots of the effect of MG132 (proteasomal

inhibitor) and NH₄Cl (lysosomal inhibitor), quantitation (from different independent experiments) of these immunoblots should be plotted.

A5: Thank you for your valuable suggestion. In response, we have quantified the effects of MG132 (proteasomal inhibitor) and NH₄Cl (lysosomal inhibitor) on TGFBR1 degradation. Statistical analyses of these immunoblots, derived from independent experiments, are now presented in the Revised Figure 4e and Supplementary Figure 4d (Below Panel).

In line with the description in our original manuscript, these quantitative results confirm that TGFBR1 degradation is mediated primarily through the proteasomal pathway. Specifically, we observed a significant elevation in TGFBR1 protein levels following treatment with MG132, whereas no substantial increase was noted with NH₄Cl in HepG2 and PLC/PRF/5 cells with stable NPC1 knockdown. These findings suggest that NPC1 promotes the TGF-β pathway by preventing proteasome-mediated TGFBR1 degradation, which has now been further supported by the statistical quantification added to the revised figures.

Fig. 4e, HepG2 cells with or without stable knockdown of NPC1 were treated with vehicle, MG132 (10 μM), or NH₄Cl (10 mM) for 12 hours. Cell lysates were subjected to immunoblot with TGFBR1 or NPC1 antibody.

Supplementary Fig. 4d, PLC/PRF/5 cells with or without stable knockdown of NPC1 were treated with vehicle, MG132 (10 μM), or NH₄Cl (10 mM) for 12 hours. Cell lysates were subjected to immunoblot with TGFBR1 or NPC1 antibody.

Data are presented as the mean ± s.e.m. of three independent experiments. Statistical significance was determined by two-tailed unpaired Student's t-test

Minor comments:

Q6: Immunoprecipitations do not imply direct binding; therefore the authors should remove the claim that NPC1 and TGFBR1 directly interact.

A6: Thank you for your suggestion. We totally agree that immunoprecipitation does not confirm direct binding between NPC1 and TGFBR1. In light of this, we have revised the manuscript to remove any claims of a direct interaction.

Q7: Authors should cite relevant work in relation to the role of NPC1 in cell migration:

PMID: 24746815, PMID: 35022465.

A7: Thank you for suggesting the inclusion of relevant citations. We have now cited the recommended studies (PMID: 24746815, PMID: 35022465) in our manuscript to appropriately reference the role of NPC1 in cell migration. We appreciate your guidance in ensuring that our manuscript is well-supported by existing literature.

Q8: Results shown in Fig 3 g should be better explained in the manuscript.

A8: Thank you for your comment. We agree that the results in Figure 3g require further explanation for clarity. To assess the cholesterol transport activity in our NPC1 constructs, we employed Filipin III staining, a widely accepted fluorescent probe that specifically binds to unesterified cholesterol in fixed cells¹¹. This method allows us to visually assess cholesterol accumulation as an indicator of impaired transport function. As shown in Figure 3g, NPC1 knockdown resulted in a significant accumulation of cholesterol within the cells, as expected. Upon re-expression of wild-type NPC1, which has intact cholesterol transport function, we observed a rescue of the cholesterol accumulation phenotype, with levels returning to normal. However, when cells were re-expressed with the P691S mutant NPC1, which is known to lack cholesterol transport activity^{12,13}, cholesterol remained accumulated, confirming that this mutant construct does not restore transport function.

Fig. 3g, NPC1-knockdown HepG2 cells with further overexpression of NPC1 or NPC1 (P691S) cells were fixed and stained with filipin to label free cholesterol accumulated in LE/Ly. Representative images from three independent experiments are shown; scale bars, 10 μ m.

Q9: In lines 304-307 the authors comment on serum lipid/lipoprotein content of *Npc1* cKO: “Furthermore, serum levels of triglyceride (TG), cholesterol (CHOL), high-density lipoprotein cholesterol (HDL-C), low-density lipoprotein cholesterol (LDL-C) and very low-density lipoprotein (VLDL) were all found to be lower in *Npc1*cKO mice compared to control mice”.

The authors should better explain these findings in relation to the NPC1-cholesterol-SREBP axis, which regulates transcription of multiple genes related to lipid metabolism and LDLR trafficking.

A9: Thank you for your insightful suggestion. In response, we conducted additional experiments to further investigate the role of NPC1 in different HCC models. Specifically, we crossed *Cre^{Alb}Npc1^{F/F}* mice with *H11-CAG-LSL-Myc* mice to generate homozygous *Npc1* knockout *Cre^{Alb}Npc1^{F/F}Myc* mice (Below Panel, Revised Fig. 7j). Liver cancer spontaneously developed in 6- to 8-week-old *Cre^{Alb}Myc* mice, and we observed that knockout of *Npc1* significantly reduced tumor sizes and weights (Below Panel, Revised Fig. 7k-m and Revised Supplementary Fig. 7k). In this MYC-driven HCC model, we also evaluated the same markers as in the DEN-CCl₄ model and

reached the same conclusion, namely, that hepatic *Npc1* knockout inhibits TGF- β signaling and MYC-driven HCC progression (Below Panel, Revised Fig.7 n, o and Revised Supplementary Fig. 7l-q).

Fig. 7j, Scheme used to establish the model of spontaneous HCC with targeted *Myc* knock-in and *Npc1* knockout in the liver.

Fig. 7k, Representative images of *Cre^{Alb}Myc* and *Cre^{Alb}Npc1^{F/F}Myc* mouse livers with HCC.

Fig. 7l, liver to body weight ratio (n = 5) in the indicated mice.

Fig. 7m, Representative H&E staining images of the indicated mouse livers. Scale bars: left panels, 2.5 mm; right panels, 500 μ m.

Supplementary Fig. 7k, liver weight of *Cre^{Alb}Myc* and *Cre^{Alb}Npc1^{F/F}Myc* mice (n = 5).

Fig. 7n, Representative IHC staining images of EpCAM, GPR78 and Cytokeratin19 of the indicated mouse livers. Scale bars, 50 μ m.

Fig. 7o, TGFBR1, p-SMAD2, SMAD2, and NPC1 protein expression in WT, *Cre^{Alb}Myc* and *Cre^{Alb}Npc1^{F/F}Myc* mouse liver tissues.

Supplementary Fig. 7l, Representative reticulin staining images of the indicated mouse livers.

Scale bars: left panels, 500 μm ; right panels, 100 μm .

Supplementary Fig. 7m, Representative IHC staining images of Ki67 in the indicated mouse livers. Scale bars: left panels, 500 μm ; right panels, 100 μm .

Supplementary Fig. 7n, Quantification of Ki67 staining in the indicated mouse livers (n = 8).

Supplementary Fig. 7o-q, Analysis of serum ALT (o), AST (p) and TBIL (q) levels in WT, *Cre^{Alb}Myc* and *Cre^{Alb}Npc1^{F/F}Myc* mice (n = 6).

Data are presented as means \pm s.e.m. Statistical significance was determined by two-tailed unpaired Student's t-test.

Our manuscript aims to highlight the novel role of NPC1 in regulating HCC progression, independent of its cholesterol transport function. However, as you noted, the NPC1-cholesterol-SREBP axis plays a well-established role in lipid metabolism.

In response to your concerns, we have reanalyzed the impact of *Npc1* knockout on lipid metabolism and the SREBP pathway in both the DEN-CCl₄ induced and MYC-driven HCC models. The data demonstrate that *Npc1* deficiency influences serum and liver lipid and modulates key proteins in the SREBP pathway, though its effects on HCC progression appear to extend beyond cholesterol metabolism (Below Panel A-L).

In the DEN-CCl₄-induced HCC model: Oil Red O staining revealed increased lipid accumulation in HCC livers, which diminished after *Npc1* knockout (Below Panel A). Liver CHOL and TG levels were elevated in HCC, with TG levels decreasing after *Npc1* knockout, while liver CHOL levels remained unchanged (Below Panel B, C). Serum CHOL and TG levels were elevated in HCC mice but significantly reduced after *Npc1* knockout (Below Panel D, E).

In the MYC-driven HCC model: Oil Red O staining revealed no significant difference across groups (Below Panel G). Liver CHOL levels increased further after *Npc1* knockout, in contrast to the DEN-CCl₄ model, where CHOL levels were unaffected (Below Panel H). Liver TG levels showed no significant difference across groups (Below Panel I). Serum TG levels were elevated in HCC but reduced after *Npc1* knockout, while serum CHOL remained unchanged (Below Panel J, K). These findings suggest that NPC1 loss has differential effects on lipid metabolism in the two HCC models, indicating variations in the regulation of cholesterol homeostasis.

To further clarify the relationship between NPC1 and lipid metabolism, we investigated the SREBP2 pathway in both models:

In the DEN-CCl₄ model: Cleaved SREBP2 was downregulated in HCC but restored after *Npc1* knockout. LDLR and HMGCS1, both SREBP2 targets, were downregulated in HCC and upregulated after *Npc1* knockout. A similar trend was observed for ABCG5 and ABCG8, which regulate cholesterol efflux to the intestinal lumen and bile ducts. HMGCR, the rate-limiting enzyme in cholesterol biosynthesis, was upregulated in HCC but did not change after *Npc1* knockout (Below Panel F).

In the MYC-driven model: SREBP2 and HMGCS1 were upregulated in HCC, with no significant change after *Npc1* knockout. LDLR and ABCG8 showed a similar pattern to the DEN-CCl₄ model, decreasing in HCC and increasing after *Npc1* knockout. Interestingly, HMGCR levels were unaffected by *Npc1* knockout in both models, suggesting post-translational regulation of this key enzyme. This observation indicates

that cholesterol biosynthesis may be regulated through multiple feedback mechanisms that fine-tune its activity independent of SREBP2-driven transcription (Below Panel L).

Taken together, our results underscore that while NPC1 plays a well-defined role in cholesterol transport and metabolism, its contribution to HCC progression extends beyond this pathway. *Npc1* deficiency alters lipid metabolism through the SREBP2 axis, but the lack of change in HMGCR in both models highlights the complexity of cholesterol regulation in HCC. Additionally, the persistent elevation of liver cholesterol in MYC-driven HCC following *Npc1* knockout further supports our hypothesis that NPC1's role in HCC is not solely reliant on its cholesterol transport function.

These findings align with recent studies suggesting that cholesterol homeostasis is tightly regulated by multiple layers of control, including transcriptional and post-translational mechanisms¹⁴. However, our work emphasizes the cholesterol-independent roles of NPC1 in HCC progression, through pathways involving TGF- β signaling.

We hope this expanded discussion provides a more nuanced explanation of the NPC1-cholesterol-SREBP axis while emphasizing our key finding that NPC1 promotes HCC independently of its cholesterol transport function, a novel aspect of our study.

Panel A-L | The impact of *Npc1* knockout on lipid metabolism and the SREBP pathway in both the DEN-CCl₄ induced and MYC-driven HCC models. **A**, Oil Red O staining in the indicated mouse livers; scale bars, 100 μ m. **B**, **C**, Analysis of liver cholesterol (CHOL) and triglyceride (TG) in indicated groups (n = 4). **D**, **E**, Analysis of serum cholesterol (TC) and triglyceride (TG) levels in normal *Npc1^{F/F}* (n = 3), *Npc1^{F/F}* (n = 4) and *Cre^{Alb}*Npc1^{F/F}** (n = 5) mice with DEN-CCl₄ induced HCC. **F**, Western blot analysis of liver tissues from the indicated mice (n = 3). **G**, Oil Red O staining in the indicated mouse livers; scale bars, 100 μ m. **H**, **I**, Analysis of liver cholesterol (CHOL) and triglyceride (TG) in indicated groups (n = 4). **J**, **K**, Analysis of serum cholesterol (TC) and triglyceride (TG) levels in WT, *Cre^{Alb}*Myc** and *Cre^{Alb}*Npc1^{F/F}*Myc** mice (n = 4). **L**, Western blot analysis of liver tissues from the indicated mice*

(n = 3). Data are presented as the mean \pm s.e.m. Statistical significance was determined by two-tailed unpaired Student's t-test.

Q10: Please check for English grammar and typos throughout the text: For example, Line 150 is missing a word in this sentence “that the critical of NPC1” (should it read “critical role”?).

A10: Thank you for pointing out the specific grammatical issue in Line 150 and for highlighting the importance of checking the entire manuscript for such errors. We have thoroughly reviewed the text to correct any grammatical mistakes and typos, including the error you mentioned. The sentence on Line 150 has been revised to “the critical role of NPC1.” We appreciate your attention to detail and have ensured that the manuscript is now more polished.

Reviewer #2

In this study, Li et al. decipher the role of NPC1, which was first identified as a potential risk marker in HCC in their previous paper using proteomics (Jiang et al., 2019). But, one recent study also has suggested the oncogenic role and prognostic value of NPC1 in HCC via regulation of autophagy (Xu et al., 2024). Here, by using in vitro and in vivo models of HCC, the authors propose that NPC1 directly interacts with TGFBR1 and prevents the binding between TGFBR1 and the SMAD7/SMURFs complex to maintain the stability of TGFBR1, thus activating TGF- β signaling in a cholesterol transport-independent manner and promoting HCC progression. This is an interesting topic, and the novelty relies on the effect and mechanism of NPC1 on cholesterol-independent regulation. Although the authors provide a large amount of experimental data with good quality, the downstream mechanism via TGF- β signaling is well recognized in HCC, and the interaction between TGF- β and SMAD7/SMURFs is also well known, which dampens the novelty of their study. Overall, the following comments should be addressed to strengthen the overall scientific soundness of the manuscript:

Q1: In Figure 2 and extended data Fig.2, authors re-induced NPC1 in HCC cells to prevent off-target effects from shRNA knockdown. The NPC1 protein level in shNPC1+NPC1 group was much higher than that in the shCTRL group, whereas most of the function studies, including cell proliferation, migration, invasion, as well as the tail vein metastasis model, showed no difference between the two groups or only had a partial rescue effect. How do the authors explain these findings? According to Figures 2a-e, it is anticipated that re-induced NPC1 should promote metastasis compared to Vector.

A1: Thank you for your insightful observation. The experiments involving cell proliferation and invasion were indeed performed later using a different batch of HCC cells than those used in the earlier migration experiments. To ensure clarity, we have included the Western blot results for this subsequent batch of cells, which indicate that

the reintroduced NPC1 expression levels in the shNPC1+NPC1 group were comparable to those observed in the shCTRL+Vector cells. These data suggest that the discrepancy between NPC1 levels and the extent of rescue observed may be attributed to differences in experimental conditions between batches.

Additionally, we have re-performed the migration and invasion assays using this new batch of cells, and the updated results are now presented in Revised Fig. 2a-e and Revised Supplementary Fig. 2a-f (Below Panel). These revised experiments show a more consistent rescue of NPC1 function across the functional assays.

Fig. 2a, Confirmation of NPC1 overexpression (NPC1), NPC1 knockdown and re-expression in HCC cells.

Fig. 2b-e, Transwell assay to examine the effect of NPC1 on HCC cell migration (**b,c**) or invasion (**d, e**) ($n = 3$ biological replicates); scale bars, 100 μ m.

Supplementary Fig. 2a, Confirmation of NPC1 knockdown and re-expression in HCC cells.

Supplementary Fig. 2e-f, Transwell assay to examine the effect of NPC1 on HCC cell migration (**e**) or invasion (**f**) ($n = 3$ biological replicates); scale bars, 100 μ m.

Data are presented as the mean \pm s.e.m. Statistical significance was determined by two-tailed unpaired Student's t-test.

Q2: Besides the metastasis model, Subcutaneous/orthotopic tumor inoculation

should be conducted to verify whether NPC1 knockdown and reinduction could promote tumor growth.

A2: Thank you for your valuable suggestion. In response, we performed a subcutaneous tumor inoculation model to investigate the effect of NPC1 knockdown and reinduction on tumor growth. As detailed in the revised manuscript, we observed that NPC1 knockdown significantly decreased tumor growth, as evidenced by reduced tumor sizes and weights (Below Panel, Revised Supplementary Fig. 2j-m). Furthermore, the reintroduction of NPC1 effectively restored tumor growth, supporting the role of NPC1 in promoting tumor development (Below Panel, Revised Supplementary Fig. 2j-m).

Supplementary Fig. 2j-m, Confirmation of NPC1 knockdown and re-expression in PLC/PRF/5 cells (j). Photographs of xenograft tumors induced by the subcutaneous inoculation of NCG mice (n = 5 mice per group) with NPC1-knockdown PLC/PRF/5 cells with further overexpression of NPC1 (k). Graphs of xenograft tumor volumes (l), and xenograft tumor weights (m). Data are presented as the mean ± s.e.m.. Statistical significance was determined by two-tailed unpaired Student's t-test.

Q3: In Figures 3a-f, please clarify the statement where TGF-β signaling downstream proteins p-SMAD2 and p-SMAD3 in NPC1-overexpressed HCC cells seemed to be only upregulated and detectable under TGF-β1 treatment. Authors also claimed that NPC1 knockdown decreased HCC cell migration capacity independent of TGF-β1 treatment. Does this mean that despite TGFBR1 increase, SMAD2/3 activation is not the crucial signaling regulated by NPC1 in driving HCC metastasis?

A3: Thank you for your insightful comment. To clarify, p-SMAD2 and p-SMAD3 in NPC1-overexpressing HCC cells are indeed detectable even without TGF-β1 stimulation; however, their activation is significantly enhanced under TGF-β1 treatment, as indicated by the longer exposure of the p-SMAD2/3 western blot bands we have now provided in Panel A (below).

Additionally, we examined the effect of NPC1 overexpression on cell migration both with and without TGF-β1 stimulation. Our results showed a significant increase in migration ability in PLC/PRF/5 cells with stable NPC1 overexpression in both groups. Notably, migration in the Vector group treated with TGF-β1 was higher compared to the NPC1-overexpressed group without TGF-β1 treatment (Revised Supplementary Fig. 3g). These findings suggest that SMAD2/3 activation is indeed a crucial pathway

regulated by NPC1 in promoting HCC metastasis.

Legend: (A) Immunoblot analysis of TGFBR1, p-SMAD2, p-SMAD3, SMAD2, SMAD3 and NPC1 expression in PLC/PRF/5 cells with NPC1 stable overexpression; right panels, longer exposure of the p-SMAD2/3 western blot bands.

Supplementary Fig. 3g, Transwell assay was performed in NPC1-overexpression PLC/PRF/5 cells with or without TGF-β1 (10 ng/mL) treatment (n = 3 biological replicates); scale bars, 100 μm. Data are presented as the mean ± s.e.m.. Statistical significance was determined by two-tailed unpaired Student’s t-test.

Q4: In the Method of cell invasion assays, NPC1 expression did influence HCC cell proliferation (extended Data Fig.2a-d) and showed a significant difference at 48 or 72 hours. Therefore, 72 hours selected for invasion assays is not convincing, and a shorter incubation period should be chosen to exclude the effects on cell proliferation.

A4: Thank you for your valuable suggestion. Following your advice, we have re-conducted both the migration and invasion assays, ensuring that a shorter incubation period (48 h) was used to minimize the potential impact of NPC1 on cell proliferation. The updated results are now presented in Revised Fig. 2d, e and Revised Supplementary Fig. 2f (below panel). Our results consistently show that NPC1 significantly promotes both migration and invasion. Additionally, the methods section has been corrected accordingly in the revised manuscript to reflect these changes.

Fig. 2d

Fig. 2e

Supplementary Fig. 2f

Fig. 2d-e, Transwell assay to examine the effect of NPC1 on indicated PLC/PRF/5 cells (**d**) or HepG2 cells (**e**) with further overexpression of NPC1 invasion (n = 3 biological replicates); scale bars, 100 μ m.

Supplementary Fig. 2f, Transwell assay to examine the effect of NPC1 on indicated MHCC-97H cell invasion (n = 3 biological replicates); scale bars, 100 μ m.

Data are presented as the mean \pm s.e.m.. Statistical significance was determined by two-tailed unpaired Student's t-test.

Q5: Have the authors detected the phospho-TGFBR1 (activated state) alteration by NPC1? If so, through what kind of mechanism? Does TGFBR1 mainly rely on phosphorylation to exert its effect on SMAD2/3? Does NPC1 only increase pan-TGFBR1, which is then phosphorylated by TGFBR2? Additionally, in Figure 6a, it is confusing that the inhibitor of TGFBR1 activation Galunisertib also inhibited pan-TGFBR1 expression level.

A5: Following your suggestion, we analyzed the levels of phospho-TGFBR1 in HCC cells with NPC1 stable knockdown. The level of phospho-TGFBR1 (T186) phosphorylation, the site is located in the GS region associated with TGFBR1 activation, was down-regulated in NPC1 knockdown HCC cells (below panel A). Based on the research in this paper, we suggest that the total protein level of TGFBR1 is reduced, and thus the phosphorylation level of TGFBR1 protein is also reduced. TGFBR1 primarily exerts its effects on SMAD2 and SMAD3 through phosphorylation. TGFBR1, upon activation by ligand binding, gets phosphorylated by TGFBR2, which then phosphorylates SMAD2/3 to activate downstream signaling. This is a well-established pathway in the TGF- β signaling mechanism. Based on the results in Figures 4 and 5 of this paper, we believe that NPC1 inhibits the binding of SMAD7/SMURFs to TGFBR1 by interacting with TGFBR1, thereby inhibiting the degradation of TGFBR1 and enhancing the overall level of TGFBR1 in cells, which can be phosphorylated by TGFBR2.

Galunisertib primarily acts by inhibiting the kinase activity of TGFBR1, preventing the receptor from phosphorylating downstream SMAD proteins. However,

as shown in the below panels B and C, and further supported by the Revised Supplementary Figure 4e, there is evidence suggesting that treatment with Galunisertib can also lead to a reduction in total TGFBR1 protein levels. The data demonstrate that after long-term TGF- β 1 treatment, the total protein level of TGFBR1 was upregulated in HCC cells, and after 16 h of Galunisertib treatment, the levels of TGFBR1, p-TGFBR1 and pSMAD2 were significantly downregulated. When cells were treated with TGF- β 1 at the same time, the downregulation of TGFBR1 protein level caused by Galunisertib treatment was more significant. This effect might be due to feedback mechanisms within the cell that respond to the inhibition of TGFBR1's activity. When TGFBR1's signaling is blocked, the cell might reduce the expression or increase the degradation of TGFBR1 to maintain homeostasis. This could be part of a broader regulatory mechanism where the inhibition of TGFBR1 leads to downregulation of the receptor itself, potentially to avoid accumulation of inactive receptors that might otherwise lead to cellular dysfunction. The decrease in TGFBR1 expression observed after Galunisertib treatment could also be due to indirect effects, such as changes in the transcriptional regulation of TGFBR1 or modulation of other signaling pathways that impact TGFBR1 stability^{15,16}.

Legend: (A) Immunoblot analysis of p-SMAD2, p-SMAD3, p-TGFBR1 (T186) and NPC1 expression in PLC/PRF/5 cells with NPC1 stable knockdown. (B) Immunoblot analysis of p-SMAD2, p-TGFBR1 (T186) and NPC1 expression in PLC/PRF/5 cells with or without TGFBR1 inhibitor LY2157299 (10 μ M) treatment. (C) Immunoblot analysis of TGFBR1, p-SMAD2, SMAD2 and NPC1 expression in PLC/PRF/5 cells treated with LY2157299 (10 μ M) combined with TGF- β 1 (10 ng/mL) for 16 h.

Supplementary Fig. 4e, PLC/PRF/5 cells with stable overexpression of NPC1 were stimulated with TGF- β 1 (10 ng/mL) for the times indicated. Levels of p-SMAD2, SMAD2, TGFBR1 and NPC1 were assayed by western blot. Data are presented as the mean \pm s.e.m. of three independent experiments. Statistical significance was determined by two-tailed unpaired Student's t-test.

Q6: Does SMAD7/SMURFs share the same binding domain with SMAD2/3 to TGFBR1, and which one has a higher intrinsic binding affinity to TGFBR1? Besides its function on enhancing the degradation of TGFBR1, SMAD7/SMURFs might competitively impede the TGFBR1-SMAD2/3 binding. By inhibiting NPC1, TGFBR1's binding domain might be more accessible to SMAD2/3. These are some questions the author should carefully demonstrate.

A6: Thank you for your insightful questions. Currently, detailed structural information on the receptor-SMAD complex is still limited. However, models of SMAD interactions have been inferred based on structural analogies to the FHA domain^{17,18}. The FHA domain is a well-characterized module that specifically binds phosphothreonine-containing peptides, and it is hypothesized that the basic patch on R-SMADs forms a recognition site for the phosphorylated GS loop of TGFBR1. This explains the specificity of receptor binding after phosphorylation. The GS domain, together with the amino-terminal kinase lobe, is predicted to serve as a docking site for the MH2 domain of R-Smads^{18,19}. Nevertheless, further structural studies are needed to fully elucidate these interactions.

Regarding binding affinity, the intrinsic differences between SMAD2 and SMAD7 with TGFBR1 remain unclear. What is well-established, however, is that SMAD7 forms a stable interaction with TGFBR1, functioning as an inhibitor at an early stage of TGF- β signaling⁵. SMAD7 competes with SMAD2/3 for binding to TGFBR1, preventing their phosphorylation and activation, thus blocking downstream signaling⁵. This competitive binding hinders the formation of the SMAD2/4 complex and subsequent nuclear translocation of SMAD2, indicating that SMAD7 may have a higher binding affinity compared to SMAD2/3 when serving as an inhibitor.

Additionally, SMAD7 plays a crucial role by recruiting E3 ubiquitin ligases such as SMURF1, and SMURF2 to TGFBR1, facilitating its ubiquitin-mediated degradation^{20,21}. Our experimental data (Below Panel, Revised Fig. 5j, k) show that NPC1 inhibition enhances the interaction of SMAD7/SMURFs with TGFBR1, suggesting that NPC1 regulates the accessibility of TGFBR1 for SMAD2/3 by modulating its interaction with SMAD7. Co-localization experiments further support these findings, demonstrating that in the context of NPC1 knockdown, the interaction between SMAD7-EGFP and TGFBR1-mCherry significantly increases (Below Panel, Revised Supplementary Fig. 5n, o).

Moreover, knockdown of SMAD7 or SMURF1/2 rescued TGFBR1 protein levels in HCC cells with stable NPC1 knockdown (Below Panel, Revised Supplementary Fig. 5p, q), supporting the idea that NPC1 regulates TGF- β signaling by influencing SMAD7's competitive binding with SMAD2/3.

Fig. 5j**Fig. 5k****Supplementary Fig. 5n****Supplementary Fig. 5o****Supplementary Fig. 5p****Supplementary Fig. 5q**
Fig. 5j-k, TGFBR1-mCherry-His stable overexpression PLC/PRF/5 (**j**) and HepG2 (**k**) cells with or without NPC1 overexpression (**j**) or knockdown (**k**) were subjected to immunoprecipitation with anti-His antibody. The lysates and immunoprecipitates were then blotted.

Supplementary Fig. 5n, PLC/PRF/5 cells stably expressing TGFBR1-mCherry-His and SMAD7-EGFP were immunostained with antibodies against LAMP1 and then imaged by Structure Illumination Microscopy to determine the colocalization among TGFBR1, SMAD7 and LAMP1 in PLC/PRF/5 cells with NPC1 stable knockdown. The images were collected through the Polar-SIM system.

Supplementary Fig. 5o, Quantification of the colocalization between TGFBR1 and SMAD7 using the Mander's coefficient in PLC/PRF/5 cells with NPC1 stable knockdown.

Supplementary Fig. 5p, Immunoblot analysis of TGFBR1, SMAD7 and NPC1 expression in NPC1-knockdown HepG2 and PLC/PRF/5 cells with or without knockdown of SMAD7.

Supplementary Fig. 5q, Immunoblot analysis of TGFBR1, SMURF1, SMURF2 and NPC1

expression in NPC1-knockdown PLC/PRF/5 cells were transfected with specific siRNAs targeting SMURF1 or SMURF2. Representative images from three independent experiments are shown; scale bars, 10 μ m. Data are presented as the mean \pm s.e.m. Statistical significance was determined by two-tailed unpaired Student's t-test.

Q7: In the DEN/CCl₄-driven mouse HCC model, the authors demonstrated that Npc1KO mice displayed fewer tumor lesions with lower serum levels of components involved in cholesterol transport, compared to control mice. However, the mechanism of NPC1-TGFBR1 axis-induced HCC metastasis illustrated is cholesterol-transport independent. Since such model induces fibrosis and inflammation-associated HCC linked to lipid dysfunction, an alternative HCC mouse model of combined hydrodynamic injection of proto-oncogenes (e.g. c-myc) and/or tumor-suppressive genes (e.g. sgp53 and keap1) may be more suitable for in vivo function validation in this context.

A7: Thank you for your insightful suggestion. In response, we conducted additional experiments to further investigate the role of NPC1 in different HCC models. Specifically, we crossed *Cre^{Alb}Npc1^{F/F}* mice with *H11-CAG-LSL-Myc* mice to generate homozygous *Npc1* knockout *Cre^{Alb}Npc1^{F/F}Myc* mice (Below Panel, Revised Fig. 7j). Liver cancer spontaneously developed in 6- to 8-week-old *Cre^{Alb}Myc* mice, and we observed that knockout of *Npc1* significantly reduced tumor sizes and weights (Below Panel, Revised Fig. 7k-m and Revised Supplementary Fig. 7k). In this MYC-driven HCC model, we also evaluated the same markers as in the DEN-CCl₄ model and reached the same conclusion, namely, that hepatic *Npc1* knockout inhibits TGF- β signaling and MYC-driven HCC progression (Below Panel, Revised Fig. 7 n, o and Revised Supplementary Fig. 7l-q).

Fig. 7j, Scheme used to establish the model of spontaneous HCC with targeted *Myc* knock-in and *Npc1* knockout in the liver.

Fig. 7k, Representative images of *Cre^{Alb}Myc* and *Cre^{Alb}Npc1^{F/F}Myc* mouse livers with HCC.

Fig. 7l, liver to body weight ratio (n = 5) in the indicated mice.

Fig. 7m, Representative H&E staining images of the indicated mouse livers. Scale bars: left panels, 2.5 mm; right panels, 500 μ m.

Supplementary Fig. 7k, liver weight of *Cre^{Alb}Myc* and *Cre^{Alb}Npc1^{F/F}Myc* mice (n = 5).

Fig. 7n, Representative IHC staining images of EpCAM, GPR78 and Cytokeratin19 of the indicated mouse livers. Scale bars, 50 μ m.

Fig. 7o, TGFBR1, p-SMAD2, SMAD2, and NPC1 protein expression in WT, *Cre^{Alb}Myc* and *Cre^{Alb}Npc1^{F/F}Myc* mouse liver tissues.

Supplementary Fig. 7l, Representative reticulin staining images of the indicated mouse livers. Scale bars: left panels, 500 μ m; right panels, 100 μ m.

Supplementary Fig. 7m, Representative IHC staining images of Ki67 in the indicated mouse livers. Scale bars: left panels, 500 μ m; right panels, 100 μ m.

Supplementary Fig. 7n, Quantification of Ki67 staining in the indicated mouse livers (n = 8).

Supplementary Fig. 7o-q. Analysis of serum ALT (o), AST (p) and TBIL (q) levels in WT, *Cre^{Alb}Myc* and *Cre^{Alb}Npc1^{F/F}Myc* mice (n = 6).

Data are presented as means \pm s.e.m. Statistical significance was determined by two-tailed unpaired Student's t-test.

Our manuscript aims to highlight the novel role of NPC1 in regulating HCC progression, independent of its cholesterol transport function. However, as you noted, the NPC1-cholesterol-SREBP axis plays a well-established role in lipid metabolism.

In response to your concerns, we have reanalyzed the impact of *Npc1* knockout on lipid metabolism and the SREBP pathway in both the DEN-CCl₄ induced and MYC-driven HCC models. The data demonstrate that *Npc1* deficiency influences serum and liver lipid and modulates key proteins in the SREBP pathway, though its effects on HCC progression appear to extend beyond cholesterol metabolism (Below Panel A-L).

In the DEN-CCl₄-induced HCC model: Oil Red O staining revealed increased lipid accumulation in HCC livers, which diminished after *Npc1* knockout (Below Panel A). Liver CHOL and TG levels were elevated in HCC, with TG levels decreasing after *Npc1* knockout, while liver CHOL levels remained unchanged (Below Panel B, C). Serum CHOL and TG levels were elevated in HCC mice but significantly reduced after *Npc1* knockout (Below Panel D, E).

In the MYC-driven HCC model: Oil Red O staining revealed no significant difference across groups (Below Panel G). Liver CHOL levels increased further after *Npc1* knockout, in contrast to the DEN-CCl₄ model, where CHOL levels were unaffected (Below Panel H). Liver TG levels showed no significant difference across groups (Below Panel I). Serum TG levels were elevated in HCC but reduced after *Npc1* knockout, while serum CHOL remained unchanged (Below Panel J, K). These findings suggest that NPC1 loss has differential effects on lipid metabolism in the two HCC models, indicating variations in the regulation of cholesterol homeostasis.

To further clarify the relationship between NPC1 and lipid metabolism, we investigated the SREBP2 pathway in both models:

In the DEN-CCl₄ model: Cleaved SREBP2 was downregulated in HCC but restored after *Npc1* knockout. LDLR and HMGCS1, both SREBP2 targets, were downregulated in HCC and upregulated after *Npc1* knockout. A similar trend was observed for ABCG5 and ABCG8, which regulate cholesterol efflux to the intestinal lumen and bile ducts. HMGCR, the rate-limiting enzyme in cholesterol biosynthesis, was upregulated in HCC but did not change after *Npc1* knockout (Below Panel F).

In the MYC-driven model: SREBP2 and HMGCS1 were upregulated in HCC, with no significant change after *Npc1* knockout. LDLR and ABCG8 showed a similar pattern to the DEN-CCl₄ model, decreasing in HCC and increasing after *Npc1* knockout. Interestingly, HMGCR levels were unaffected by *Npc1* knockout in both models, suggesting post-translational regulation of this key enzyme. This observation indicates that cholesterol biosynthesis may be regulated through multiple feedback mechanisms that fine-tune its activity independent of SREBP2-driven transcription (Below Panel L).

Taken together, our results underscore that while NPC1 plays a well-defined role in cholesterol transport and metabolism, its contribution to HCC progression extends

beyond this pathway. *Npc1* deficiency alters lipid metabolism through the SREBP2 axis, but the lack of change in HMGCR in both models highlights the complexity of cholesterol regulation in HCC. Additionally, the persistent elevation of liver cholesterol in MYC-driven HCC following *Npc1* knockout further supports our hypothesis that NPC1's role in HCC is not solely reliant on its cholesterol transport function.

These findings align with recent studies suggesting that cholesterol homeostasis is tightly regulated by multiple layers of control, including transcriptional and post-translational mechanisms¹⁴. However, our work emphasizes the cholesterol-independent roles of NPC1 in HCC progression, through pathways involving TGF- β signaling.

We hope this expanded discussion provides a more nuanced explanation of the NPC1-cholesterol-SREBP axis while emphasizing our key finding that NPC1 promotes HCC independently of its cholesterol transport function, a novel aspect of our study.

Panel A-L | The impact of *Npc1* knockout on lipid metabolism and the SREBP pathway in both the DEN-CCl₄ induced and MYC-driven HCC models. **A**, Oil Red O staining in the indicated mouse livers; scale bars, 100 μ m. **B**, **C**, Analysis of liver cholesterol (CHOL) and triglyceride (TG) in indicated groups (n = 4). **D**, **E**, Analysis of serum cholesterol (TC) and triglyceride (TG) levels in normal *Npc1^{F/F}* (n = 3), *Npc1^{F/F}* (n = 4) and *Cre^{Alb}*Npc1^{F/F}** (n = 5) mice with DEN-CCl₄ induced HCC. **F**, Western blot analysis of liver tissues from the indicated mice (n = 3). **G**, Oil Red O staining in the indicated mouse livers; scale bars, 100 μ m. **H**, **I**, Analysis of liver cholesterol (CHOL) and triglyceride (TG) in indicated groups (n = 4). **J**, **K**, Analysis of serum cholesterol (TC) and triglyceride (TG) levels in WT, *Cre^{Alb}*Myc** and *Cre^{Alb}*Npc1^{F/F}**Myc* mice (n = 4). **L**, Western blot analysis of liver tissues from the indicated mice*

(n = 3). Data are presented as the mean ± s.e.m. Statistical significance was determined by two-tailed unpaired Student's t-test.

Q8: The effect of NPC1 mainly depends on TGFBR1 expression. So does it mean that if an HCC patient with tumor cells lacked TGFBR1, NPC1 would not contribute to tumor progression? The authors should first validate this hypothesis by overexpressing NPC1 in TGFBR1 knockout HCC cells and compare their function to TGFBR1-WT cells. Then, in their clinical cohorts, the authors should demonstrate whether NPC1's prognostic value is only present in TGFBR1-co-expressing HCC patients, but not in TGFBR1-low and negative patients. Also, in their spontaneous HCC model, each mouse has a different expression of TGFBR1 on tumor cells. Thus, it is important to stratify the mouse tumor samples into TGFBR1-high and TGFBR1-low groups to evaluate the tumor size/nodule number/IHC/WB, as shown in Figures 7e-o.

A8: Thank you very much for your constructive suggestions. In response, we performed additional experiments, including proliferation and migration assays, as well as a subcutaneous mouse model, to further investigate your concerns. We generated PLC/PRF/5 cells with stable NPC1 overexpression and TGFBR1 knockdown, comparing their functional behavior to TGFBR1-WT cells. Our results showed that TGFBR1 knockdown significantly suppressed the migration of PLC/PRF/5 cells with NPC1 overexpression, while it had no significant effect on cell proliferation or tumor growth in xenograft models (Below Panel, Revised Supplementary Fig. 6d-i). These findings underscore the role of TGFBR1 in NPC1-mediated HCC cell migration, while NPC1 may exert effects through additional pathways independent of TGFBR1 to influence tumor growth and proliferation²².

Supplementary Fig. 6d

Supplementary Fig. 6e

Supplementary Fig. 6f

Supplementary Fig. 6g

Supplementary Fig. 6h

Supplementary Fig. 6i

Supplementary Fig. 6d, Immunoblot analysis of TGFBR1 and NPC1 expression in

PLC/PRF/5 cells with stable overexpression or knockdown of indicated genes.

Supplementary Fig. 6e, Transwell assay was performed in cells related to (d) (n = 3 biological replicates); scale bars, 100 μ m.

Supplementary Fig. 6f, Cell growth curves were measured in indicated cells (n = 3 biological replicates).

Supplementary Fig. 6g-i, Photographs of xenograft tumors induced by the subcutaneous inoculation of NCG mice (n = 5 mice per group) in indicated groups (g). Graphs of xenograft tumor volumes (h), and xenograft tumor weights (i). Data are presented as the mean \pm s.e.m.. Statistical significance was determined by two-tailed unpaired Student's t-test.

We acknowledge that TGFBR1 expression plays an important role in NPC1-mediated promotion of HCC progression. To investigate whether NPC1's prognostic value is dependent on TGFBR1 levels, we aimed to stratify patients based on TGFBR1 expression in two proteomic cohorts (Jiang's cohort and Gao's cohort). Unfortunately, TGFBR1 was only identified in six patients in Jiang's cohort, and TGFBR1 was undetected in Gao's cohort, limiting our ability to perform the stratified analysis in these datasets.

To overcome this limitation, we conducted immunohistochemistry (IHC) analysis for TGFBR1 in our tissue microarray (TMA) cohort. We classified patients with both NPC1 and TGFBR1 intensity greater than or equal to 6 as the high group, and the rest as the low group. Kaplan-Meier survival analysis revealed that patients in the NPC1 and TGFBR1 high group had significantly better overall survival compared to those in the low group (Below Panel A). Then, we found a significant positive correlation between NPC1 and TGFBR1 protein levels ($P < 0.0001$, $R^2 = 0.5028$), indicating that high NPC1 expression is positively associated with TGFBR1 levels (Below Panel, Revised Fig. 6a-c).

Fig. 6a

Fig. 6b

Fig. 6c

Legend: (A) The Kaplan-Meier overall survival curves of individuals with NPC1 and TGFBR1 intensity greater than or equal to 6 as the high group (n=115), and the rest as the low group (n=171).

Fig. 6a, Representative images from IHC staining of NPC1 and TGFBR1 in HCC tissues (n = 286); scale bar, 50 μ m.

Fig. 6b, The Pearson correlation analysis between NPC1 level and TGFBR1 level in HCC tissues.

Fig. 6c, The analysis of TGFBR1 IHC score in HCC tissues with low (n = 174) or high (n = 112) NPC1 level. In the box plots, the middle bar represents the median, and the box represents the interquartile range; bars extend to 1.5 \times the interquartile range. Statistical significance was determined by Pearson correlation test (b), Mann–Whitney U test (c,) or log-rank test (A).

Finally, we analyzed liver tissue from 25 DEN-CCl₄ induced spontaneous HCC mouse models using Western blot for TGFBR1 expression. We ranked the samples based on TGFBR1 expression and divided them into two groups: TGFBR1-low (n=12) and TGFBR1-high (n=13). We then analyzed tumor size and number of nodules. Although we observed a trend towards larger tumors and more nodules in the TGFBR1-high group, this difference did not reach statistical significance, likely due to the limited number of animals and the relatively small variation in TGFBR1 expression across the different mouse tumor samples in the presence of NPC1. (Below Panel, B-D).

Legend: (B) NPC1 protein expression in *Npc1*^{F/F} mouse HCC tissues (n = 25). (C, D) Numbers of nodules per liver (C) and diameters of the largest nodules (D) of the indicated group. Data are presented as the mean \pm s.e.m. Statistical significance was determined by two-tailed unpaired Student's t-test.

Q9: Animal studies: please confirm if any subcutaneous mouse model was used as described in Methods. In addition, the authors claimed in the Discussion section that there has been no specific NPC1 inhibitor to test its therapeutic efficacy on HCC in mice until now. AAV8 therapy specifically targeting hepatic NPC1 in an orthotopic mouse model should be performed to validate its therapeutic potential. More importantly, the authors should compare the effects of NPC1 inhibition and TGFBR1 inhibition in vitro and in vivo, including migration/invasion assays, subcutaneous/orthotopic tumor inoculation, and tail-vein lung metastasis model.

A9: Thank you for your insightful suggestion. We performed a subcutaneous tumor inoculation experiment to answer your Q2、 Q8 and subsequent questions. We have conducted additional experiments to address your concerns regarding the role of NPC1 and TGFBR1 in HCC progression. Following your suggestion, we explored the therapeutic potential of targeting NPC1 in a MYC-driven HCC mouse model. Mice were administered AAV8-shNPC1 or a control AAV8-shCTRL via tail vein injection. After four weeks, we observed a significant antitumor effect in the AAV8-shNPC1 treated group, as evidenced by reduced tumor size and number (Below Panel, Revised Supplementary Fig. 7t-x). The knockdown efficiency of NPC1 was confirmed by Western blot analysis (Below Panel, Revised Supplementary Fig. 7y).

Supplementary Fig. 7t, Schematic view of the treatment plan using AAV serotype 8 (AAV8).

Supplementary Fig. 7u, Representative liver images of the indicated group.

Supplementary Fig. 7v-w, liver weight (v) and liver to body weight ratio (w) of the indicated group (n = 6).

Supplementary Fig. 7x, Representative H&E staining images of the indicated mouse livers. Scale bars: left panels, 2.5 mm; right panels, 500 μ m.

Supplementary Fig. 7y, NPC1 protein expression in the indicated mouse tissues. Data are presented as the mean \pm s.e.m. (v-w). Statistical significance was determined by two-tailed unpaired Student's t-test (v-w). Data are presented as the mean \pm s.e.m. Statistical significance was determined by two-tailed unpaired Student's t-test.

Additionally, we utilized an inducible knockout of *Npc1* in a DEN-CC14-induced HCC model using tamoxifen (TAM)-inducible Cre recombinase (ERT2-Cre). The deletion of *Npc1* for an eight-week period led to a significant reduction in tumor burden, as demonstrated by decreased tumor number and size, as well as a lower liver-to-body weight ratio (Below Panel, Revised Fig. 7p-s, u and Supplementary Fig. 7r, s). Histological analysis revealed regressed tumor morphology in *Npc1*-knockout livers compared to controls (Below Panel, Revised Fig. 7t). These findings strongly suggest that NPC1 is a promising therapeutic target for HCC.

Fig. 7p**Fig. 7q****Fig. 7u****Fig. 7r****Fig. 7s****Supplementary Fig. 7r****Supplementary Fig. 7s****Fig. 7t**
Fig. 7p, Schematic diagram of tamoxifen-induced liver-specific *Npc1* knockout mouse generation and the treatment plan in the DEN-CCl₄-induced HCC model.

Fig. 7q, Representative liver images of the indicated group.

Fig. 7r-s, liver to body weight ratio (**r**), numbers of nodules per liver (**s**) of *Npc1*^{F/F} mice treated with coil (CO) (n = 5), *CreERT2*^{Alb}*Npc1*^{F/F} mice treated with coil or tamoxifen (TAM) (n = 8).

Supplementary Fig. 7r-s, liver weight (**r**), diameters of the largest nodules (**s**) of *Npc1*^{F/F} mice treated with coil (CO) (n = 5), *CreERT2*^{Alb}*Npc1*^{F/F} mice treated with coil (CO) or tamoxifen (TAM) (n = 8).

Fig. 7t, Representative H&E staining images of the indicated mouse livers. Scale bars: left panels, 2.5 mm; right panels, 500 μm.

Fig. 7u, NPC1 protein expression in the indicated mouse tissues.

Data are presented as means ± s.e.m. Statistical significance was determined by two-tailed unpaired Student's t-test.

To further validate your suggestion, we compared the effects of NPC1 and TGFBR1 inhibition both in vitro and in vivo. Migration assays demonstrated that knockdown of either NPC1 or TGFBR1 significantly inhibited the migration of PLC/RPF/5 and MHCC-97H cells (Below Panel, Revised Supplementary Fig. 6d, e, j, k). Additionally, while NPC1 knockdown suppressed cell proliferation and tumor growth in PLC/RPF/5 cells, TGFBR1 knockdown did not significantly impact these phenotypes (Below Panel, Revised Supplementary Fig. 6f-i)

Lastly, using an NCG mouse tail vein metastasis model, we assessed the metastatic

potential of MHCC-97H cells with either NPC1 or TGFB β 1 knockdown. Both knockdowns significantly reduced metastasis, underscoring the importance of these proteins in HCC progression (Below Panel, Revised Supplementary Fig. 6l, m).

We hope that these additional experiments and findings sufficiently address your concerns and highlight the therapeutic relevance of targeting NPC1 in HCC.

Supplementary Fig. 6l

Supplementary Fig. 6m

Supplementary Fig. 6d, Immunoblot analysis of TGFBR1 and NPC1 expression in PLC/PRF/5 cells with stable overexpression or knockdown of indicated genes.

Supplementary Fig. 6e, Transwell assay was performed in cells related to (d) (n = 3 biological replicates); scale bars, 100 μm.

Supplementary Fig. 6f, Cell growth curves were measured in indicated cells (n = 3 biological replicates).

Supplementary Fig. 6g-i, Photographs of xenograft tumors induced by the subcutaneous inoculation of NCG mice (n = 5 mice per group) in indicated groups (g). Graphs of xenograft tumor volumes (h), and xenograft tumor weights (i).

Supplementary Fig. 6j, Immunoblot analysis of TGFBR1 and NPC1 expression in MHCC-97H cells with stable overexpression or knockdown of indicated genes.

Supplementary Fig. 6k, Transwell assay was performed in cells related to (j) (n = 3 biological replicates); scale bars, 100 μm.

Supplementary Fig. 6l-m, 1×10^6 Luciferase-expressing MHCC-97H cells were injected into NCG mice by tail vein. The mice were euthanized 8 weeks later by a cervical dislocation. Representative H&E staining images of lung tissues are shown; scale bars, 1mm; insets: twofold magnification; scale bars, 500 μm (l). The incidence of lung metastasis in mice (m) (n = 6 mice per group). Data are presented as the mean \pm s.e.m.. Statistical significance was determined by two-tailed unpaired Student's t-test.

Q10: Careful grammar check has to be carried out since the manuscript has multiple grammatical errors.

A10: Thank you for your feedback regarding the grammatical issues in our manuscript. We have carefully reviewed and revised the manuscript to correct any grammatical errors and improve overall readability. We appreciate your attention to detail and believe that the changes have enhanced the clarity and quality of our submission.

Reviewer #3

The authors argue that NPC1 controls TGFbetaRI (TbRI) stability in a cholesterol-independent manner is promoting hepatocellular carcinoma progression. They have used conventional methods to knock down NPC1 in vitro and in vivo and show that

NPC1 promotes tumor progression in vitro and in vivo. They propose that NPC1 is causing increased stability of TβRI by preventing Smad7 to recruit SMURFs, which targets TβRI for degradation.

The presented data are interesting and novel, but I have some concerns about their data and the conclusions they draw.

Q1: On line 101 the authors claim that NPC1 “directly interacts with” TβRI. To claim this, they should show direct interactions between the two proteins.

A1: Thank you for your suggestion. We totally agree that immunoprecipitation does not confirm direct binding between NPC1 and TGFβR1. In light of this, we have revised the manuscript to remove any claims of a direct interaction.

Q2: If NPC1 recruits Smad7 and SMURFs to the TβRI, they should show that this is dependent on TGFβ treatment of cells. From the literature we know that “Smad7 recruits the HECT type of E3 ubiquitin ligases, Smurf1 and Smurf2. It binds to Smurfs in the nucleus and translocates into the cytoplasm in response to TGF-β and recruits the ubiquitin ligases to the activated type I receptor ALK5/TβRI, leading to the degradation of the receptor through the proteasomal pathway.” Therefore, it would be of scientific interest to know in which subcellular localization the TβRI-NPC1 complex is located. The complex is shown by immunofluorescence in Figure 5c.

A2: Thank you very much for your insightful suggestion. In response, we conducted experiments to investigate the subcellular localization of the NPC1-TGFβR1 complex and its role in modulating SMAD7 and SMURFs interactions with TGFβR1.

As demonstrated in Revised Figure 5c and Revised Supplementary Fig 5b-f (Below Panel), our results show that NPC1 colocalizes with TGFβR1 in LAMP1-positive vesicles, indicating that this complex is likely localized in late endosomes or lysosomes.

Fig. 5c

Supplementary Fig. 5b

Supplementary Fig. 5c

Supplementary Fig. 5d

Supplementary Fig. 5e

Supplementary Fig. 5f

Fig. 5c, PLC/PRF/5 cells stably overexpressing TGFBR1-mCherry-His and NPC1-HA were immunostained with antibodies against HA and LAMP1 to determine the colocalization among TGFBR1, NPC1 and LAMP1 in PLC/PRF/5 cells. Representative images from three independent experiments are shown; scale bars, 10 μ m.

Supplementary Fig. 5b, HepG2 cells stably expressing TGFBR1-mCherry-His were immunostained with antibodies against NPC1 and LAMP1 to determine the colocalization among TGFBR1, NPC1 and LAMP1 in HepG2 cells.

Supplementary Fig. 5c, HepG2 cells stably expressing TGFBR1-mCherry-His and Caveolin-1 (CAV1)-Myc were immunostained with antibodies against Myc and LAMP1 to determine the colocalization among TGFBR1, CAV1 and LAMP1 in HepG2 cells.

Supplementary Fig. 5d-e, HepG2 cells stably expressing TGFBR1-mCherry-His were immunostained with antibodies against RAB5(d)/RAB11(e) and LAMP1 to determine the colocalization among TGFBR1, RAB5/ RAB11 and LAMP1 in HepG2 cells.

Supplementary Fig. 5f, Quantification of the colocalization between these four proteins (LAMP1, CAV1, RAB5 and RAB11) and TGFBR1 using the Mander's coefficient.

Representative images from three independent experiments are shown; scale bars, 10 μ m. Data are presented as the mean \pm s.e.m. Statistical significance was determined by two-tailed unpaired Student's t-test.

Further supporting this, our data in Figure 5j and 5k (Below Panel) show that inhibition of NPC1 significantly enhances the binding of SMAD7 and SMURFs to TGFBR1. This observation suggests that NPC1 may regulate TGF- β signaling by modulating the interaction between SMAD7 and TGFBR1. Specifically, in the absence of NPC1, SMAD7 and SMURFs may more readily associate with TGFBR1, promoting its ubiquitination and subsequent degradation via the proteasomal pathway.

Fig. 5j-k, TGFBR1-mCherry-His stable overexpression PLC/PRF/5 (**j**) and HepG2 (**k**) cells with or without NPC1 overexpression (**j**) or knockdown (**k**) were subjected to immunoprecipitation with anti-His antibody. The lysates and immunoprecipitates were then blotted.

Moreover, the colocalization results of SMAD7-EGFP and TGFBR1-mCherry, shown in the Revised Supplementary Figure 5n, o (Below Panel), further validate these findings. We observed that in NPC1 knockdown cells, the localization of SMAD7-EGFP in the cytoplasm significantly increases and colocalizes with TGFBR1-mCherry. This supports the hypothesis that NPC1 influences the stability of TGFBR1 by modulating its interaction with SMAD7.

Extended Fig. 5n

Extended Fig. 5o

Supplementary Fig. 5n, PLC/PRF/5 cells stably expressing TGFBR1-mCherry-His and SMAD7-EGFP were immunostained with antibodies against LAMP1 and then imaged by Structure Illumination Microscopy to determine the colocalization among TGFBR1, SMAD7 and LAMP1 in PLC/PRF/5 cells with NPC1 stable knockdown. The images were collected through the Polar-SIM system.

Supplementary Fig. 5o, Quantification of the colocalization between TGFBR1 and SMAD7 using the Mander's coefficient in PLC/PRF/5 cells with NPC1 stable knockdown. Representative images from three independent experiments are shown; scale bars, 10 μ m. Data are presented as the mean \pm s.e.m. Statistical significance was determined by two-tailed unpaired Student's t-test.

Additionally, we found that knockdown of SMAD7 or inhibition of SMURF1/2 using siRNA could rescue TGFBR1 protein levels in HCC cells with stable NPC1 knockdown (Below Panel, Revised Supplementary Fig. 5p, q). This result suggests that the degradation of TGFBR1 in NPC1 knockdown cells is mediated, at least in part, by the increased recruitment of SMAD7 and SMURFs to TGFBR1.

Supplementary Fig. 5p, Immunoblot analysis of TGFBR1, SMAD7 and NPC1 expression in NPC1-knockdown HepG2 and PLC/PRF/5 cells with or without knockdown of SMAD7.

Supplementary Fig. 5q, Immunoblot analysis of TGFBR1, SMURF1, SMURF2 and NPC1 expression in NPC1-knockdown PLC/PRF/5 cells were transfected with specific siRNAs targeting SMURF1 or SMURF2.

Importantly, all these experiments were conducted without exogenous TGF- β stimulation. Previous studies indicate that SMAD7 can stably interact with TGFBR1 or SMURF2 even in the absence of exogenous ligand stimulation, with TGF- β potentially enhancing this interaction. Given that SMAD7 acts as an inhibitor in the early stages of TGF- β signaling⁵, our findings further suggest that NPC1 inhibits the interaction between TGFBR1 and SMAD7, thereby promoting TGFBR1 stability at a very early stage in the TGF- β signaling pathway, even without TGF- β 1 stimulation.

Concerns/Questions:

Q3: Is the Tbr1-NPC1 complex localized in caveola?

A3: Thank you for your insightful question. To address your concern, we performed additional experiments to investigate the subcellular localization of the TGFBR1-NPC1 complex. Our results indicate that the TGFBR1-NPC1 complex colocalizes with the lysosomal marker LAMP1, rather than with the caveolae marker caveolin-1, as shown in the Revised Figure 5c and the revised Supplementary Figure 5b-f (Below Panel).

While literature has reported that TGFBR1 can localize to caveolae and that NPC1 may colocalize with caveolin-1 under certain conditions¹⁻⁴, our findings suggest that in HCC cells, the TGFBR1-NPC1 complex is primarily localized within lysosomes, rather

than caveolae.

Fig. 5c

Supplementary Fig. 5c

Supplementary Fig. 5e

Supplementary Fig. 5b

Supplementary Fig. 5d

Supplementary Fig. 5f

Fig. 5c, PLC/PRF/5 cells stably overexpressing TGFBR1-mCherry-His and NPC1-HA were immunostained with antibodies against HA and LAMP1 to determine the colocalization among TGFBR1, NPC1 and LAMP1 in PLC/PRF/5 cells. Representative images from three independent experiments are shown; scale bars, 10 μ m.

Supplementary Fig. 5b, HepG2 cells stably expressing TGFBR1-mCherry-His were immunostained with antibodies against NPC1 and LAMP1 to determine the colocalization among TGFBR1, NPC1 and LAMP1 in HepG2 cells.

Supplementary Fig. 5c, HepG2 cells stably expressing TGFBR1-mCherry-His and Caveolin-1 (CAV1)-Myc were immunostained with antibodies against Myc and LAMP1 to determine the colocalization among TGFBR1, CAV1 and LAMP1 in HepG2 cells.

Supplementary Fig. 5d-e, HepG2 cells stably expressing TGFBR1-mCherry-His were immunostained with antibodies against RAB5(d)/RAB11(e) and LAMP1 to determine the colocalization among TGFBR1, RAB5/ RAB11 and LAMP1 in HepG2 cells.

Supplementary Fig. 5f, Quantification of the colocalization between these four proteins (LAMP1, CAV1, RAB5 and RAB11) and TGFBR1 using the Mander's coefficient.

Representative images from three independent experiments are shown; scale bars, 10 μ m. Data are presented as the mean \pm s.e.m. Statistical significance was determined by two-tailed

unpaired Student's t-test.

Q4: would a mutant NPC1 lacking 692-854 region, not recruit TbRI to this subcellular localization?

A4: Thank you for your insightful question. We investigated whether the mutant NPC1 lacking the 692-854 region would still recruit TGFBR1 to lysosomes. Our experiments, as shown in Below Panel A, B, demonstrate that in PLC (TGFBR1-mCherry) cells where NPC1 is knockdown and then reintroduced with NPC1 (Δ 692-854), TGFBR1 still localizes to lysosomes, similar to the wild-type NPC1. Additionally, knockdown of NPC1 did not affect the lysosomal localization of TGFBR1 (Below Panel, Revised Supplementary Fig. 5b, g, i).

Supplementary Fig. 5b

Supplementary Fig. 5g

Supplementary Fig. 5i

Legend: (A) NPC1-knockdown PLC/PRF/5 cells stably expressing TGFBR1-mCherry-His with further overexpression of NPC1 or NPC1 (Δ 692-854) were immunostained with antibodies against LAMP1 and then imaged by Structure Illumination Microscopy to determine the colocalization between TGFBR1 and LAMP1 in indicated cells. The images were collected through the Polar-SIM system (Airy Technology Co., Ltd., China). A 640 nm laser were used to excite mouse Alexa Fluor 633 with 2DSIM modality. The SIM reconstruction process was conducted using the Airy-SIM software with pre-processing (Dark) or post precessing (MRA); Scale bar, 10 μ m. (B) Quantification of the colocalization between TGFBR1 and LAMP1 using the Mander's coefficient.

Supplementary Fig. 5b, HepG2 cells stably expressing TGFBR1-mCherry-His were immunostained with antibodies against NPC1 and LAMP1 to determine the colocalization among TGFBR1, NPC1 and LAMP1 in HepG2 cells.

Supplementary Fig. 5g, NPC1-knockdown HepG2 cells stably expressing TGFBR1-mCherry-

His were immunostained with antibodies against NPC1 and LAMP1 to determine the colocalization among TGFBR1, NPC1 and LAMP1 in HepG2 cells.

Supplementary Fig. 5i, Quantification of the colocalization between TGFBR1 and LAMP1 using the Mander's coefficient in HepG2 cells with NPC1 stable knockdown

Representative images from three independent experiments are shown; scale bars, 10 μ m. Data are presented as the mean \pm s.e.m. Statistical significance was determined by two-tailed unpaired Student's t-test.

Importantly, our fluorescence co-localization results reveal that while the wild-type NPC1 is localized to lysosomes, the NPC1 mutant lacking the 692-854 region is distributed in the cytoplasm and does not co-localize with LAMP1 (Below Panel C, D). This finding indicates that the 692-854 region of NPC1 is crucial for its lysosomal localization. These results collectively suggest that the 692-854 region of NPC1 is essential for its interaction with TGFBR1 in lysosomes.

Legend: (C) Confirmation of NPC1 knockdown and then re-expression NPC1-HA or NPC1 (Δ692-854)-HA in PLC/PRF/5 cells. (D) immunofluorescence staining with anti-HA and anti-LAMP1 antibody to determine the colocalization between NPC1 (or NPC1 (Δ692-854)) and LAMP1 proteins in indicated PLC/PRF/5 cells; scale bars, 5 μ m. Data are presented as the mean \pm s.e.m. Statistical significance was determined by two-tailed unpaired Student's t-test.

Q5: what happens with this subcellular complex (TbRI-NPC1) in presence of TGFbeta stimulation of cells? Will treatment with Galunisertib change the colocalisation of TbRI and NPC1?

A5: Thank you for your insightful question regarding the effects of TGF- β stimulation and Galunisertib treatment on the subcellular localization of the TGFBR1-NPC1 complex.

To investigate this, we performed experiments in PLC (TGFBR1-mCherry-His)

cells, comparing the co-localization of TGFBR1, NPC1, and LAMP1 with or without TGF- β 1 stimulation. Using anti-NPC1 and anti-LAMP1 antibodies, we quantified the co-localization of TGFBR1 and LAMP1 in both conditions. The results demonstrate that TGFBR1-NPC1 remains localized to lysosomes irrespective of TGF- β 1 stimulation, with no significant change in the co-localization coefficient between TGFBR1 and LAMP1 in the stimulated versus unstimulated groups (Below Panel A, B).

We also examined the effects of Galunisertib treatment in PLC (TGFBR1-mCherry-His) cells. Using the same antibodies, we quantified TGFBR1, NPC1, and LAMP1 co-localization with and without Galunisertib treatment. The results indicate that TGFBR1-NPC1 continues to localize to lysosomes regardless of Galunisertib treatment, and there was no observable impact of Galunisertib on the localization of the TGFBR1-NPC1 complex (Below Panel A, B).

These findings suggest that the localization of the TGFBR1-NPC1 complex to lysosomes is stable under both TGF- β 1 stimulation and Galunisertib treatment.

Legend: (A) After treatment with TGF- β 1(10 ng/mL) for 30min or TGFBR1 inhibitor LY2157299 (10 μ M) for 16 h, PLC/PRF/5 cells stably expressing TGFBR1-mCherry were immunostained with antibodies against LAMP1 and NPC1 and then imaged by Structure Illumination Microscopy to determine the colocalization among TGFBR1, LAMP1 and NPC1 in indicated cells. The images were collected through the Polar-SIM system (Airy Technology Co., Ltd., China). A 488 nm (or 640) laser were used to excite rabbit Alexa Fluor 488(or mouse Alexa Fluor 633) with 2DSIM modality. The SIM reconstruction process was conducted using the Airy-SIM software with pre-processing (Dark) or post precessing (MRA); Scale bar, 10 μ m. (B) Quantification of the colocalization between TGFBR1 and LAMP1 using the Mander's coefficient. Data are presented as the mean \pm s.e.m. Statistical significance was determined by two-tailed unpaired Student's t-test.

Q6: If NPC1 control TbRI protein stability, it would be interesting to know if NPC1 and TbRI are co-expressed in HCC tissue sections. This can be investigated with immunohistochemistry.

Does high expression of TbRI also correlate with poor prognosis for patients with HCC?

A6: Thank you for your insightful suggestion. In line with your recommendation, we examined the correlation between NPC1 and TGFBR1 protein levels in 286 pairs of human HCC tissue samples using immunohistochemistry (IHC). Our analysis revealed a significant positive correlation between NPC1 and TGFBR1 expression ($P < 0.0001$, $R^2 = 0.5028$) (Below Panel, Revised Fig. 6a-c). To further explore this relationship, we performed immunofluorescence staining on HCC tissues with antibodies specific to NPC1 and TGFBR1. The results confirmed that high NPC1 expression is positively correlated with TGFBR1 levels in these tissues (Below Panel, Revised Fig. 6d-f).

Fig. 6a, Representative images from IHC staining of NPC1 and TGFBR1 in HCC tissues (n = 286); scale bar, 50 μ m.

Fig. 6b, The Pearson correlation analysis between NPC1 level and TGFBR1 level in HCC tissues.

Fig. 6c, The analysis of TGFBR1 IHC score in HCC tissues with low (n = 174) or high (n = 112) NPC1 level.

Fig. 6d, Representative images from Immunofluorescence staining of NPC1 and TGFBR1 in

HCC tissues (n = 10); scale bar, 20 μ m.

Fig. 6e, The Pearson correlation analysis between NPC1 positive area and TGFBR1 positive area in HCC tissues (n = 50). These multiplexed IF staining were performed on ten HCC tissue sections from HCC patients, qualifying an average of five regions per sample.

Fig. 6f, The analysis of the percentages of TGFBR1 positive area in HCC tissues with low (n = 13) or high (n = 37) NPC1 level. In the box plots, the middle bar represents the median, and the box represents the interquartile range; bars extend to 1.5 \times the interquartile range. Statistical significance was determined by Pearson correlation test (**b,e**), or Mann–Whitney U test (**c,f**).

Additionally, we conducted a Kaplan-Meier survival analysis to determine the prognostic significance of TGFBR1 expression in HCC patients. Our analysis showed that TGFBR1 expression levels were not significantly associated with overall survival (OS) or disease-free survival (DFS) (Below Panel A, B). The result suggest that TGFBR1's role in HCC progression is more complex. High TGFBR1 expression alone may not directly predict poor prognosis in patients with HCC, as the impact of TGF- β signaling could depend on additional factors such as tumor stage or the presence of mutations in downstream effectors²³.

Legend: (A, B) Kaplan-Meier overall survival (A) and disease-free survival (B) curves of individuals with high or low TGFBR1 expression. Statistical significance was determined by log-rank test.

Q7: Discussion line 346; “Galunisertib is the only TGFbeta pathway inhibitor currently under clinical investigation in HCC patients” this should be revised; as far as I know is the effect of Galunisertib no longer investigated in clinical trials.

A7: Thank you for pointing out the outdated information regarding Galunisertib. We agree that the statement in the discussion requires revision. As current clinical trials no longer investigate Galunisertib for HCC patients, we will update the text accordingly to reflect this. We propose revising the sentence to: "Galunisertib, a potent TGFBR1 inhibitor, has been investigated in clinical trials for HCC²⁴⁻²⁹." We appreciate your attention to this detail and have made the necessary correction to ensure our discussion is up to date.

Reference

- 1 Chen, Y. G. Endocytic regulation of TGF-beta signaling. *Cell Res* **19**, 58-70, doi:10.1038/cr.2008.315 (2009).

- 2 He, K. *et al.* Internalization of the TGF-beta type I receptor into caveolin-1 and EEA1
double-positive early endosomes. *Cell Res* **25**, 738-752, doi:10.1038/cr.2015.60 (2015).
- 3 Garver, W. S., Heidenreich, R. A., Erickson, R. P., Thomas, M. A. & Wilson, J. M. Localization
of the murine Niemann-Pick C1 protein to two distinct intracellular compartments. *J Lipid*
Res **41**, 673-687 (2000).
- 4 Jelinek, D., Heidenreich, R. A., Orlando, R. A. & Garver, W. S. The Niemann-Pick C1 and
caveolin-1 proteins interact to modulate efflux of low density lipoprotein-derived
cholesterol from late endocytic compartments. *J Mol Biochem* **3**, 14-26 (2014).
- 5 Hayashi, H. *et al.* The MAD-related protein Smad7 associates with the TGFbeta receptor
and functions as an antagonist of TGFbeta signaling. *Cell* **89**, 1165-1173,
doi:10.1016/s0092-8674(00)80303-7 (1997).
- 6 Di Guglielmo, G. M., Le Roy, C., Goodfellow, A. F. & Wrana, J. L. Distinct endocytic pathways
regulate TGF-beta receptor signalling and turnover. *Nat Cell Biol* **5**, 410-421,
doi:10.1038/ncb975 (2003).
- 7 Mitchell, H., Choudhury, A., Pagano, R. E. & Leof, E. B. Ligand-dependent and -
independent transforming growth factor-beta receptor recycling regulated by clathrin-
mediated endocytosis and Rab11. *Mol Biol Cell* **15**, 4166-4178, doi:10.1091/mbc.e04-03-
0245 (2004).
- 8 Miller, D. S. J. *et al.* The Dynamics of TGF-beta Signaling Are Dictated by Receptor
Trafficking via the ESCRT Machinery. *Cell Rep* **25**, 1841-1855 e1845,
doi:10.1016/j.celrep.2018.10.056 (2018).
- 9 Lu, F. *et al.* Identification of NPC1 as the target of U18666A, an inhibitor of lysosomal
cholesterol export and Ebola infection. *Elife* **4**, doi:10.7554/eLife.12177 (2015).
- 10 Cenedella, R. J. Cholesterol synthesis inhibitor U18666A and the role of sterol metabolism
and trafficking in numerous pathophysiological processes. *Lipids* **44**, 477-487,
doi:10.1007/s11745-009-3305-7 (2009).
- 11 Castanho, M. A., Coutinho, A. & Prieto, M. J. Absorption and fluorescence spectra of
polyene antibiotics in the presence of cholesterol. *J Biol Chem* **267**, 204-209 (1992).
- 12 Watari, H. *et al.* Mutations in the leucine zipper motif and sterol-sensing domain inactivate
the Niemann-Pick C1 glycoprotein. *J Biol Chem* **274**, 21861-21866,
doi:10.1074/jbc.274.31.21861 (1999).
- 13 Millard, E. E. *et al.* The sterol-sensing domain of the Niemann-Pick C1 (NPC1) protein
regulates trafficking of low density lipoprotein cholesterol. *J Biol Chem* **280**, 28581-28590,
doi:10.1074/jbc.M414024200 (2005).
- 14 Luo, J., Yang, H. & Song, B. L. Mechanisms and regulation of cholesterol homeostasis. *Nat*
Rev Mol Cell Biol **21**, 225-245, doi:10.1038/s41580-019-0190-7 (2020).
- 15 Yan, X., Xiong, X. & Chen, Y. G. Feedback regulation of TGF-beta signaling. *Acta Biochim*
Biophys Sin (Shanghai) **50**, 37-50, doi:10.1093/abbs/gmx129 (2018).
- 16 Huang, F. & Chen, Y. G. Regulation of TGF-beta receptor activity. *Cell Biosci* **2**, 9,
doi:10.1186/2045-3701-2-9 (2012).
- 17 Durocher, D. *et al.* The molecular basis of FHA domain:phosphopeptide binding specificity
and implications for phospho-dependent signaling mechanisms. *Mol Cell* **6**, 1169-1182,
doi:10.1016/s1097-2765(00)00114-3 (2000).
- 18 Huse, M. *et al.* The TGF beta receptor activation process: an inhibitor- to substrate-

- binding switch. *Mol Cell* **8**, 671-682, doi:10.1016/s1097-2765(01)00332-x (2001).
- 19 Chaikuad, A. & Bullock, A. N. Structural Basis of Intracellular TGF-beta Signaling: Receptors and Smads. *Cold Spring Harb Perspect Biol* **8**, doi:10.1101/cshperspect.a022111 (2016).
- 20 Kavsak, P. *et al.* Smad7 binds to Smurf2 to form an E3 ubiquitin ligase that targets the TGF beta receptor for degradation. *Mol Cell* **6**, 1365-1375, doi:10.1016/s1097-2765(00)00134-9 (2000).
- 21 Ebisawa, T. *et al.* Smurf1 interacts with transforming growth factor-beta type I receptor through Smad7 and induces receptor degradation. *J Biol Chem* **276**, 12477-12480, doi:10.1074/jbc.C100008200 (2001).
- 22 Xu, J., Chen, F., Zhu, W. & Zhang, W. NPC1 promotes autophagy with tumor promotion and acts as a prognostic model for hepatocellular carcinoma. *Gene* **897**, 148050, doi:10.1016/j.gene.2023.148050 (2024).
- 23 Seoane, J. & Gomis, R. R. TGF-beta Family Signaling in Tumor Suppression and Cancer Progression. *Cold Spring Harb Perspect Biol* **9**, doi:10.1101/cshperspect.a022277 (2017).
- 24 Derynck, R., Turley, S. J. & Akhurst, R. J. TGFbeta biology in cancer progression and immunotherapy. *Nat Rev Clin Oncol* **18**, 9-34, doi:10.1038/s41571-020-0403-1 (2021).
- 25 Ungerleider, N., Han, C., Zhang, J., Yao, L. & Wu, T. TGFbeta signaling confers sorafenib resistance via induction of multiple RTKs in hepatocellular carcinoma cells. *Mol Carcinog* **56**, 1302-1311, doi:10.1002/mc.22592 (2017).
- 26 Kelley, R. K. *et al.* A Phase 2 Study of Galunisertib (TGF-beta1 Receptor Type I Inhibitor) and Sorafenib in Patients With Advanced Hepatocellular Carcinoma. *Clin Transl Gastroenterol* **10**, e00056, doi:10.14309/ctg.0000000000000056 (2019).
- 27 Ikeda, M. *et al.* Phase 1b study of galunisertib in combination with gemcitabine in Japanese patients with metastatic or locally advanced pancreatic cancer. *Cancer Chemother Pharmacol* **79**, 1169-1177, doi:10.1007/s00280-017-3313-x (2017).
- 28 Ikeda, M. *et al.* A phase 1b study of transforming growth factor-beta receptor I inhibitor galunisertib in combination with sorafenib in Japanese patients with unresectable hepatocellular carcinoma. *Invest New Drugs* **37**, 118-126, doi:10.1007/s10637-018-0636-3 (2019).
- 29 Harding, J. J. *et al.* Phase 1b study of galunisertib and ramucirumab in patients with advanced hepatocellular carcinoma. *Cancer Med* **10**, 3059-3067, doi:10.1002/cam4.3880 (2021).

Dear Editor,

Thank you for giving us the opportunity to respond to the reviewers' comments point by point, on our manuscript (ID: NCOMMS-23-61307B) entitled "**NPC1 controls TGFBR1 stability in a cholesterol transport-independent manner and promotes hepatocellular carcinoma progression**" for consideration for publication in *Nature Communications*. We would like to express our sincere gratitude for the invaluable feedback provided by the reviewers and editors. Their detailed and constructive comments have greatly contributed to improving the quality of the manuscript. We have carefully addressed all remaining comments from the reviewer, which are described below, along with our responses to each comment. We hope the responses meet the expectations of the reviewers and editors, and we look forward to your feedback.

Reviewer #1

In this revised version of their manuscript, Li and colleagues have addressed basically all comments that I had previously raised. Moreover, they have also addressed point by point other reviewer's important concerns.

In spite of this, I believe that over-expression of TGFBR1 probably leads to over-interpretation of some of their conclusions regarding trafficking of this receptor. As the authors note, previous studies have shown TGFBR1 internalization independent of TGF β 1 stimulation, however, the recycling compartment has nothing to do with lysosomes. Here, in their rebuttal letter, the authors show lysosomal localization of TGFBR1 regardless of TGF β 1-mediated stimulation. This is somewhat unexpected, since I would expect an increase in the amount of receptor molecules that reach the lysosomal compartment (after internalization), where NPC1 resides. The fact that over-expressed TGFBR1 localizes in lysosomes at steady state (in unstimulated conditions) makes me wonder whether this could be an artefactual effect of over-expression. If a good anti-TGFBR1 antibody exists for immunofluorescence experiments, labeling of endogenous TGFBR1 in both unstimulated and TGF β 1-stimulated cells would solve this issue. Otherwise, the authors should consider this potential pitfall in their over-expression system and clearly reflect this issue in the manuscript.

Finally, in their Revised Supplementary Fig. 4e, I noted that TGFBR1 protein levels increase from 4 to 24 h of TGF β 1 stimulation. Is this expected? Others have shown lysosomal-mediated TGFBR1 downregulation after TGF β 1 stimulation (PMID: 12717440 ; PMID: 30428352). Please provide some comment to this.

Altogether, I acknowledge the author's efforts to address the multiple concerns raised by myself and the rest of reviewers. Overall, the manuscript has increased its quality. Publication is recommended after addressing these two minor comments stated above.

Re: We sincerely thank the reviewer for the insightful feedback on the potential impact of TGFBR1 over-expression on receptor trafficking and localization.

To address this concern, we verified that the TGFBR1-mCherry-His construct was expressed at a reasonable level in PLC/PRF/5 cells, as shown in the Panel A. Importantly, we observed that the overexpression of TGFBR1-mCherry-His had only a minor effect on signal transduction, as indicated by the data presented in Panels A-C. These findings are consistent with previous reports¹⁻⁴, suggesting that the overexpression of TGFBR1 in our system does not significantly alter its normal cellular behavior.

To further validate our observations, we conducted experiments using three commercially available antibodies targeting endogenous TGFBR1. In TGFBR1 knockdown and control PLC/PRF/5 cells, we performed both Western blot (WB) validation and immunofluorescence (IF) analysis. In the WB validation (Below Panel D), only antibody ab235578 showed reliable and specific detection of TGFBR1. However, in the IF experiments (Panel E), none of the three antibodies demonstrated satisfactory specificity for endogenous TGFBR1. As a result, we were unable to perform high-quality IF staining of endogenous TGFBR1 using these commercial antibodies. We have also revised the manuscript to acknowledge this limitation and to emphasize the potential influence of over-expression on our results.

Therefore, given the limitations of the currently available antibodies for IF detection of endogenous TGFBR1, we opted to use the TGFBR1-mCherry-His construct for detailed localization studies. We hope these clarifications and additional data address your concerns regarding the specificity of TGFBR1 detection in our system.

A, Confirmation of TGFBR1 overexpression in PLC/PRF/5 cells.
B, PLC/PRF/5 cells stably expressing TGFBR1-mCherry-His with or without TGF- β 1 (10 ng/mL) treatment for 30 min were immunostained with antibodies against p-SMAD2 and then imaged by spinning disk confocal microscope. Scale bar, 20 μ m. Quantification of p-SMAD2 f
C, Immunoblot analysis of TGFBR1, p-SMAD2 and SMAD2 expression in TGFBR1-knockdown PLC/PRF/5 cells with further overexpression of TGFBR1-mCherry-His.
D, Validation of three TGFBR1 antibodies by western blot analysis in PLC/PRF/5 cells with TGFBR1 knockdown.
E, Validation of three TGFBR1 antibodies by Immunofluorescence analysis in PLC/PRF/5

s
c
e
n
c

cells with TGFBR1 knockdown. Scale bar, 15 μ m.

In response to the second concern about TGFBR1 levels increasing between 4 to 24 hours of TGF- β 1 stimulation, we acknowledge that this observation was initially unexpected. In our experiments, we observed that TGFBR1 protein levels were indeed upregulated in hepatocellular carcinoma (HCC) cells after extended TGF- β 1 treatment.

For example, the study referenced by the reviewer (PMID: 30428352) showed no significant increase in total TGFBR1 levels under similar conditions. In contrast, the findings in PMID: 12717440 emphasize the complexity of TGFBR1 regulation, as receptor dynamics can be influenced by cellular context and specific experimental conditions.

Our results suggest that TGFBR1 might be regulated differently in HCC cells, potentially due to specific cellular or environmental factors. This discrepancy highlights the need for further investigation into the context-dependent behavior of TGFBR1, particularly in cancer models.

Reviewer #2

The authors have satisfactorily addressed my comments.

Re: We are grateful to Reviewer #2 for the positive feedback. We are pleased to hear that our previous revisions have satisfactorily addressed your comments. Thank you for your thorough review and constructive suggestions, which have helped improve the quality of our manuscript.

Reviewer #3

The revised manuscript have included data from additional experiments as suggested. I am satisfied with the new data as presented in the revised manuscript and have no further questions.

Re: We sincerely thank Reviewer #3 for the positive feedback and for recognizing the additional experiments we have included in the revised manuscript. We are pleased that the new data has satisfactorily addressed your previous concerns. Thank you for your thorough review and valuable input throughout this process.

Reference

- 1 He, K. *et al.* Internalization of the TGF-beta type I receptor into caveolin-1 and EEA1 double-positive early endosomes. *Cell Res* **25**, 738-752, doi:10.1038/cr.2015.60 (2015).
- 2 Di Guglielmo, G. M., Le Roy, C., Goodfellow, A. F. & Wrana, J. L. Distinct endocytic pathways regulate TGF-beta receptor signalling and turnover. *Nat Cell Biol* **5**, 410-421, doi:10.1038/ncb975 (2003).
- 3 Mitchell, H., Choudhury, A., Pagano, R. E. & Leof, E. B. Ligand-dependent and

-independent transforming growth factor-beta receptor recycling regulated by clathrin-mediated endocytosis and Rab11. *Mol Biol Cell* **15**, 4166-4178, doi:10.1091/mbc.e04-03-0245 (2004).

- 4 Hayes, S., Chawla, A. & Corvera, S. TGF beta receptor internalization into EEA1-enriched early endosomes: role in signaling to Smad2. *J Cell Biol* **158**, 1239-1249, doi:10.1083/jcb.200204088 (2002).